# Reactant friendly hydrogen evolution interface based on di-anionic MoS$_2$ surface

Zhaoyan Luo[1,2,5], Hao Zhang[3,5], Yuqi Yang[3,5], Xian Wang[1,2,5], Yang Li[1,2,5], Zhao Jin[1], Zheng Jiang [3,4✉], Changpeng Liu[1], Wei Xing[1] & Junjie Ge [1✉]

Engineering the reaction interface to preferentially attract reactants to inner Helmholtz plane is highly desirable for kinetic advancement of most electro-catalysis processes, including hydrogen evolution reaction (HER). This, however, has rarely been achieved due to the inherent complexity for precise surface manipulation down to molecule level. Here, we build a MoS$_2$ di-anionic surface with controlled molecular substitution of S sites by –OH. We confirm the –OH group endows the interface with reactant dragging functionality, through forming strong non-covalent hydrogen bonding to the reactants (hydronium ions or water). The well-conditioned surface, in conjunction with activated sulfur atoms (by heteroatom metal doping) as active sites, giving rise to up-to-date the lowest over potential and highest intrinsic activity among all the MoS$_2$ based catalysts. The di-anion surface created in this study, with atomic mixing of active sites and reactant dragging functionalities, represents a effective di-functional interface for boosted kinetic performance.

[1] State Key Laboratory of Electroanalytical Chemistry, Jilin Province Key Laboratory of Low Carbon Chemical Power, Changchun Institute of Applied Chemistry, Chinese Academy of Sciences, 130022 Changchun, China. [2] University of Science and Technology of China, 230026 Anhui, China. [3] Shanghai Advanced Research Institute, Chinese Academy of Sciences, 201800 Shanghai, China. [4] Shanghai Synchrotron Radiation Facility, Zhangjiang National Lab, Chinese Academy of Sciences, 201204 Shanghai, China. [5] These authors contributed equally: Zhaoyan Luo, Hao Zhang, Yuqi Yang. ✉email: jiangzheng@sinap.ac.cn; gejj@ciac.ac.cn

The hydrogen production from electrochemical water splitting, which enables the energy cycling between electricity and hydrogen, constitutes the cornerstone of the sustainable hydrogen economy. Hydrogen evolution reaction (HER) electrocatalysts that are fast in kinetics, low in energy consumption, and cost-effective in nature were intensively searched, among which $MoS_2$ has emerged as a promising candidate[1–3]. The overall kinetic performance (in Volmer-Heyrovsky mechanism) of the $MoS_2$ electrode at fixed potential ($E$) is shown in Eq. 1[4,5] and is apparently governed by: first, the energetic interaction between atomic hydrogen and the surface site ($\Delta G_{H*}$); second, the reactant (hydronium ions in acid and water in alkaline medium) concentration. Most of the recent endeavors were paid on advancing the $MoS_2$ catalytic efficiency through the former, i.e., the orbital overlap and chemical interactions between adsorbates and the surface sites[6,7], including our recent work on Pd-doped $MoS_2$[8]. However, the latter reactant concentration part is rarely studied, which calls for extreme attention. Notably, this concentration is by no means the bulky value, but rather the one in the vicinity of the electrode. Therefore, it relies heavily on the property of the electrode materials at the interface and thereby partially determines the final catalytic behavior[9,10]. For instance, an electrode can be simultaneously of optimized $\Delta G$ for intermediate species, meanwhile expressing poor reactivity due to the unfriendly interface (double layer) structure that pushing away or inhibit the movement of reactants[11].

$$j = -\frac{2Fk_2^0[H^+]^{1-\alpha}\exp\left(\frac{-\alpha FE_{RHE}+(1-\beta)\Delta G_H}{RT}\right)}{1 + \frac{k_{-1}^0}{k_1^0}\exp\left(\frac{FE_{RHE}+\Delta G_H}{RT}\right) + \frac{k_2^0}{k_1^0}\exp\left(\frac{\Delta G_H}{RT}\right)}, \qquad (1)$$

For the HER reaction, water molecules/hydronium ions in bulky solution are attracting each other through the non-covalent hydrogen bonding, with stabilization energy at 20–40 kJ mol$^{-1}$. The lack of such bonding between the hydronium ions/water molecules and the electrode surface would hinder them from accessing the inner Helmholtz plane (IHP), if no additional interactions (such as the hydrogen under potential deposition at Pt surface) are formed[11,12]. Therefore, a well-conditioned surface that preferentially attracts the hydronium ions/water molecules is essentially required for HER kinetic advancement. However, such interface engineering is largely neglected to date, thus leaving unpredictable values uncovered.

The oxygen-containing species can act as perfect ligands for surface engineering, under the consideration that these groups can form hydrogen bonding with hydronium ions/water molecules and attract these reactants to the surface. Previous literature have already revealed that the substitution of sulfur by oxygen species occur at certain circumstances. Specially, Levente Tapasztó et al.[13] reported unambiguously that oxygen atoms can spontaneously incorporate into the basal plane of $MoS_2$ single layers through substitutional oxidation, when subjected to long-term ambient exposure. Alexander Weber-Bargioni et al.[14] also identified the oxygen substitution on the sulfur sites in other monolayer transition metal dichalcogenides. However, it is noted that the oxidation of single layer basal plane already exhibits a high kinetic barriers of ~1.0 eV, thus leading to a slow oxidation kinetics with a timescale of months at room temperature. Harsh oxidation processes, on the contrary, overcome the kinetic barrier at the expense of over oxidation and destroying the original $MoS_2$ crystal lattice. Therefore, the controlled oxygen substitution is highly desirable. Meanwhile, introducing appropriate oxygen species (such as –OH) is highly important for interface engineering to preferentially attract the hydronium ions/water, however, to the best of our knowledge, this area hasn't been explored yet and therefore deserves special attention.

Here, we build a hydrogen evolution di-anionic surface on $MoS_2$ material to control its catalytic activity. Specifically, sulfur anions are electronically activated by heteroatom metal doping (Pd and Ru) to acquire optimized hydrogen adsorption energy. Meanwhile, –OH anions molecularly replace S sites at the interface in a controllable manner can create a reactant benign interface. The merits of the di-anion interface with –OH anion doping are: first, –OH functional groups attracts hydronium ions and water molecules closer to the inner Helmholtz plane (IHP) through hydrogen bonding, thus contributing to a reactant friendly interface; second, –OH sites work in conjunction with adjacent metal sites (M–OH) to split water in alkaline medium, thus enormously boosts the HER catalytic behavior. We show that the HER activity of the final catalyst exhibits highest kinetic performance exceeding the existed $MoS_2$ based material in both acidic and alkaline environments.

## Results and discussion

**Design for the di-anionic $MoS_{2-x}(OH)_y$.** We initiate by conceiving –OH functional group as a perfect ligand for surface engineering, under the consideration that surface –OH groups can form hydrogen bonding with hydronium ions/water molecules and attract these reactants to the surface. Structurally, however, –OH introduction towards forming di-anionic surface without altering the phase structure is a grand challenge[15–17].

Fundamentally, a reaction needs to be both thermodynamically and kinetically favorable in order to occur at a distinctive rate. We design to circumvent these two challenges through a sequential element substitution strategy: first, thermodynamically, we dope $MoS_2$ with minor amount of Ru to make the substitution of S by –OH energetically favorable. The density functional theory (DFT) calculations reveal that the Ru-S bond energy (0.92 eV) is 0.87 eV lower than the Mo-S bond (1.79 eV), thus is more prone to form adjacent sulfur vacancies (Supplementary Fig. 1). Meanwhile, if Ru bonded defects sites are formed, they readily capture the nucleophilic –OH species owing to their thermodynamic favorable formation energies of −4.01 ~ −4.32 eV compared to other species (oxygen atom at −2.32 ~ −2.71 eV and $O_2$ at −1.02 ~ −1.41 eV) (Supplementary Fig. 2). Experimentally, the phenomenon that Ru is highly affinitive to OH has long been recognized and utilized in catalysis for a variety of reactions such as methanol oxidation and alkaline hydrogen oxidation(in PtRu), where Ru–OH is used as the co-catalyst for CO removal/water dissociation[18]. Secomd, kinetically, we trigger the Ru–S bond substitution by Ru–OH via surface redox reaction with Pd[8]. The Pd atomic doping, which occurs spontaneously at the interface to substitute Mo sites, kills two birds with one stone: (a) facilitates the formation of SVs at the energetically more favorable Ru adjacent sites, thus triggering further –OH anchoring to form stable Ru–OH bond (Supplementary Figs. 3, 4); (b) activates the Pd bonded sulfur atoms to exhibit optimal $\Delta G_{H*}$, as we have demonstrated recently[8]. Through this technique, it is expected that both $\Delta G_H$ and the concentration terms in Eq. 1 can be optimized via building a di-anionic surface.

**Structural confirmation of the di-anionic $MoS_{2-x}(OH)_y$.** Figure 1a and Supplementary Fig. 5 illustrated the synthetic procedures for the two-step doping strategy. In the first step, Ru-doped molybdenum disulfide (Ru-$MoS_2$) was synthesized via a typical one-pot hydrothermal synthesis technique (see Methods for the details). Ru-doped $MoS_2$ catalysts with Ru mass loadings of 1–15% were synthesized through tuning the Mo:Ru ratio in the precursors. X-ray diffraction (XRD), transmission electron microscopy (TEM) and scanning electron microscopy (SEM) patterns show the maintenance of the $MoS_2$ morphological and

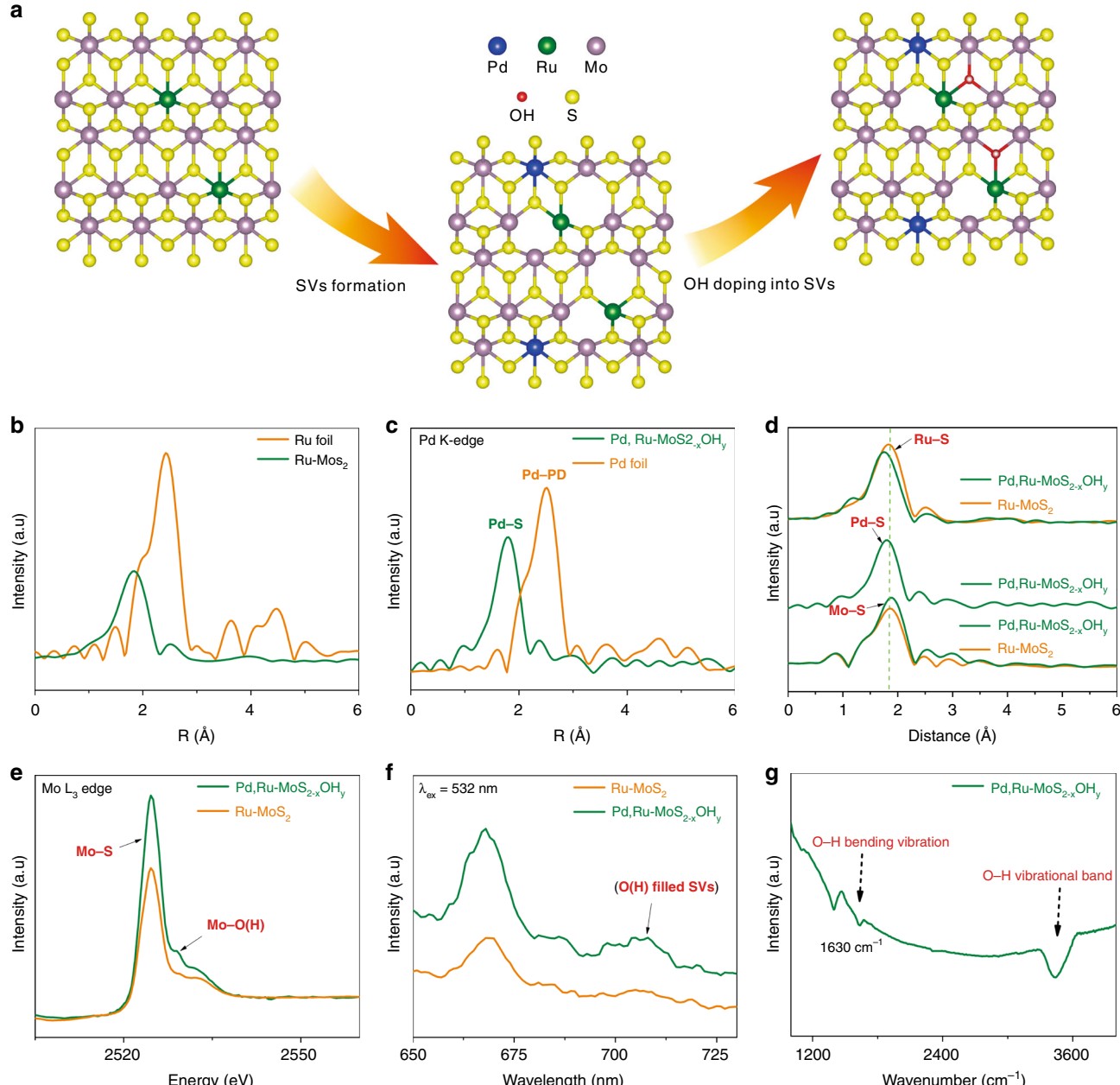

**Fig. 1 Schematics for the O filled S vacancies (SVs).** **a** Schematics for the synthesis strategy of Pd, Ru dual-doped $MoS_{2-x}OH_y$ phase. Blue, yellow, purple, green and red spheres represent Pd, S, Mo, Ru, and O atoms, respectively. **b** Fourier transform of the $k^2$-weighted Ru K-edge EXAFS spectra. **c** Fourier transform of the $k^2$-weighted Pd K-edge of EXAFS spectra. **d** Fourier transform of the $k^2$-weighted Pd K-edge, Ru K-edge and Mo K-edge EXAFS spectra. **e** Mo $L_3$-edge XANES spectra of Ru-$MoS_2$ and Pd, Ru-$MoS_{2-x}OH_y$. **f** Photoluminescence spectra of Ru-$MoS_2$ and Pd, Ru-$MoS_{2-x}OH_y$. **g** FTIR spectra of Pd, Ru-$MoS_{2-x}OH_y$.

crystal structure after Ru doping, until Ru doping content reaches 15% (Supplementary Figs. 6–8). Extended X-ray absorption fine structures (EXAFS) were measured to elucidate the Ru local bonding environment and the occupation sites in $MoS_2$, with 2.5% Ru–$MoS_2$ taken as a representative. The lack of Ru-Ru scattering path at 2.44 Å of Ru–$MoS_2$ in comparison with Ru foil verifies the atomic doping status of Ru. The Fourier transform (FT) of the $k^2$-weighted Ru K-edge (Fig. 1b and Supplementary Fig. 9a) shows a notable peak at 1.75 Å, corresponding to a first shell Ru–S scattering path. The fitting result shows a Ru–S coordination number (CN) of 6.2 and Ru–S bond length at 2.33 Å (Supplementary Table 1), suggesting Ru atoms are saturatedly coordinated by S[3].

In the second step, Pd was introduced to Ru–$MoS_2$ through a spontaneous interfacial redox strategy[8], which is the key step for the formation of Ru adjacent SVs and the following –OH anchoring. The structure of the final material (Pd, Ru–$MoS_{2-x}OH_y$) was evaluated by SEM, TEM, XRD, Raman and sub-angstrom resolution aberration-corrected HAADF-STEM microscopy. SEM and TEM images (Supplementary Figs. 10, 11) show that Pd, Ru–$MoS_{2-x}OH_y$ well retains the morphology of Ru–$MoS_2$ after Pd doping, where rose-like two-dimensional nanosheets are observed in both samples. Meanwhile, no nanoparticles or large clusters are observed in TEM image, thus ruling out the possibility of forming phase segregated palladium sulfide compounds, being consistent with the XRD patterns (Supplementary Fig. 12a). Raman spectra

(Supplementary Fig. 12b-c) of Pd, Ru–MoS$_{2-x}$OH$_y$ show similar characteristics like the Ru–MoS$_2$. Furthermore, sub-angstrom resolution high angle annular dark field-scanning transmission electron microscopy (HAADF-STEM) images and the selected area electron diffraction (SAED) pattern (Supplementary Fig. 13) demonstrate the crystalline structure and the atomic dispersion of Pd in the final sample. Combining the above results together, it is suggested that the structure of the catalysts is well retained after Pd doping, where no obvious structure disorder occurs. The local chemical structure of Pd and Ru is further probed by EXAFS. The best fitting of $k^2$-weighted Pd K-edge FT spectrum (Fig. 1c and Supplementary Fig. 14; Supplementary Table 2) shows similar profile and fitting parameters (Pd-S bond distance at 2.31 Å and CN = 3.9) to that of Mo (Mo–S bond distance at 2.30 Å and CN = 4.5) in MoS$_2$ (Supplementary Fig. 15; Supplementary Table 3), suggests that Pd substitute Mo or Ru atoms in Ru–MoS$_2$. Further evaluation of the final reactant solution (Supplementary Table. 4) by inductively coupled plasma mass spectrometry (ICP-MS) excludes the possibility of Pd to displace Ru sites, thus validating the atomic substitution of Mo by Pd. Of particular interesting is the Ru EXAFS spectrum (Fig. 1d) after Pd doping, where the nearest-neighbor FT Ru K-edge peak shows obvious shift towards the lower-R position, indicative the emerging of new bond with shorter scattering path[19]. The best fitting analyses clearly show that the path at 2.07 Å is satisfactorily interpreted as Ru–O(H) (the detailed structure of O containing species is not solved here, thus is denoted as –O(H)) contribution. The least-square EXAFS fitting analysis infer the Ru–S and Ru–O(H) coordination numbers of 4.5 (bond length 2.33 Å) and 1(bond length 2.07 Å), respectively (Supplementary Fig. 9b; Supplementary Table 1). These results clearly demonstrate the incorporation of –O(H) to the sites adjacent to Ru. The decrease in overall Ru coordination number from 6.2 to 5.5 after Pd incorporation corroborates the net creation of Ru adjacent S vacancies in spite of the partial –O(H) refilling, in consistent with our DFT calculations (Supplementary Fig. 16 and Supplementary Discussion 1). If –O(H) sites are to replace the S atoms, additional Mo–O(H) should be noticed since –O(H) is to bound with two other adjacent Mo atoms. This is validated by the Mo K-edge EXAFS (Supplementary Fig. 17 and Table 5), where the coordination numbers of Mo–O(H) is found to be 0.3. Meanwhile, through X-ray absorption near-edge spectra (XANES) investigation, the presence of Mo–O(H) is directly confirmed with a shoulder peak observed at 2527 eV in the Mo L$_3$-edge (Fig. 1e). It is noted that only when Pd and Ru are co-doped through our technique results in the incorporation of the –O(H) into the MoS$_2$ substrate. The counterparts MoS$_2$ substrates, Ru–MoS$_2$, and Pd–MoS$_2$ were all examined through XAS, where no symbolic signals for –O(H) introduction were noticed.

The necessity of Pd, Ru dual doping for –O(H) incorporation is further verified by XPS, Raman and photoluminescence (PL) spectra. In Pd, Ru-MoS$_{2-x}$(OH)$_y$, O 1$s$ peak corresponding to the binding energy of lattice oxygen is clearly observable at 530.5 eV (Supplementary Fig. 18a) in XPS spectra[20]. On the contrary, the Ru–MoS$_2$ (Supplementary Fig. 18b) and the Pd–MoS$_2$ (Supplementary Fig. 18c) sample show the absence of the corresponding peak. In Raman spectra (Supplementary Fig. 19), a new signal around 283 cm$^{-1}$ is observed, attributable to B 2 g mode of Mo–O(H) bonds[13]. In PL spectra (Fig. 1f), an obvious peak emerging at 710 nm (1.75 eV) is associated with the –O(H) filled S vacancies[13], which again only appears when MoS$_2$ is double doped by Pd and Ru. To here, it is clear that Ru and Pd are both necessary for the –O(H) group introduction, in which the former makes the reaction thermodynamically favorable and the latter triggers the reaction kinetically via surface redox reaction.

We carried out the Fourier Transform infrared spectroscopy (FTIR) spectra and the proton solid-state nuclear magnetic resonance

($^1$H SS-NMR) to further identify the oxygen-containing species in the catalysts. FTIR spectrum (Fig. 1g) of the Pd, Ru–MoS$_{2-x}$OH$_y$ sample shows a strong signal of O–H vibrational bands, with a broad peak at 3000–3500 cm$^{-1}$ signifies the O–H bond stretching vibration and a peak at 1630 cm$^{-1}$ corresponds to O–H bending vibration[21,22]. On the contrary, the MoS$_2$, Ru–MoS$_2$, and the Pd–MoS$_2$ sample show the absence of OH signature (Supplementary Fig. 20). Additionally, the $^1$H NMR spectrum (Supplementary Fig. 21 and Supplementary discussion 2) of the Pd, Ru–MoS$_{2-x}$OH$_y$ catalyst shows a peak at 2.802 ppm, which could match a hydrogen atom bonded to an oxygen atom[23]. The counterparts MoS$_2$ substrates, Ru–MoS$_2$, and Pd–MoS$_2$ were all examined through $^1$H SS-NMR spectrum, where no symbolic signals for –OH introduction were noticed. These results confirm our claim that the controlled molecular substitution of S by –OH sites was achieved by a sequential element substitution strategy. Thus, the final sample is unambiguously verified to represent a di-anionic MoS$_{2-x}$(OH)$_y$ structure.

**HER performance evaluation.** Having established the successful creation of the di-anionic surface with molecular substitution of S sites by –OH, denoted as Pd, Ru–MoS$_{2-x}$(OH)$_y$, we turned to evaluate the final HER catalytic behavior of these different catalysts, started in acidic medium (Fig. 2a and Supplementary Figs. 22, 23). First, pristine MoS$_2$ shows an over potential of 10 mA cm$^{-2}$ ($\eta$@10 mA cm$^{-2}$) at 355 mV, consistent with the values reported in literature[24,25]. Second, Pd–MoS$_2$ and Ru–MoS$_2$ exhibit much higher activity than MoS$_2$ catalyst, respectively reaching over potentials of 10 mA cm$^{-2}$ ($\eta$@10 mA cm$^{-2}$) at 201–128 mV (0.1–1 wt% Pd) and 170–140 mV (1–5 wt% Ru), mainly ascribable to the activation of in-plane S atoms[8,26] (Supplementary Fig. 24). Third, the di-anionic Pd, Ru-MoS$_{2-x}$OH$_y$ catalysts show even higher catalytic activities, especially in high current density region. Specifically, Pd, Ru-MoS$_{2-x}$OH$_y$ (Ru: 2.5%, Pd: 0.5%) shows low overpotential of 45 and 93 mV to achieve 10 and 100 mA cm$^{-2}$ current density, respectively. This value, to the best of our knowledge, is the highest performance ever reported for MoS$_2$-based catalysts in acidic media[7,13,27–30] (Supplementary Table. 6). Even more, supporting Pd, Ru–MoS$_{2-x}$OH$_y$ catalyst on rGO substrate reinforces the catalytic activity to approach or even surpass that of Pt catalyst in high current density region (Supplementary Fig. 25). The superb activity of the Pd, Ru–MoS$_{2-x}$OH$_y$ catalyst is likewise evidenced by its exchange current density ($i_0$), turnover frequency (TOF) (Supplementary Fig. 26; Table 7, Supplementary Note 1), and charge transfer resistance (Rct) derived from electrochemical impedance spectroscopy (EIS) (Supplementary Fig. 27; Table 8). Although the $i_0$ and TOF of Pd, Ru–Mo$_{2-x}$OH$_y$ is still below that of Pt[31,32], it is better than those of the best-characterized MoS$_2$-based materials[6,33].

The explicit role of –OH introduction in facilitating the HER catalytic activity is further verified by tuning the Pd content to control the –OH surface concentration. As monitored by XPS spectra (Fig. 2b and Supplementary Fig. 28; Table 9), the –OH content increases with the increase in Pd content, suggesting the essential role of Pd in introducing –OH functional group. We further plotted the current density ($J$) achieved at $\eta = 140$ mV as a function of Pd content in Pd, Ru–MoS$_{2-x}$OH$_y$ (In Fig. 2c and Supplementary Fig. 29; Table 10). In all these samples, the content of Ru is fixed at 2.5 wt%, representing $J$ of 7.58 mA cm$^{-2}$. The Pd–MoS$_2$ sample with differed Pd content was also plotted to offset the contribution from Pd. Fascinatingly, with the increase of –OH content in Pd, Ru–MoS$_{2-x}$OH$_y$, the performance gap between the two catalysts becomes more distinct. Specifically, the currents $J$ (39.07–245.66 mA cm$^{-2}$) of Pd, Ru–MoS$_{2-x}$OH$_y$ (Pd doping content 0.1–1 wt%) outperforms the superimposed currents of the counterpart Pd–MoS$_2$ (0.1–1 wt%) samples

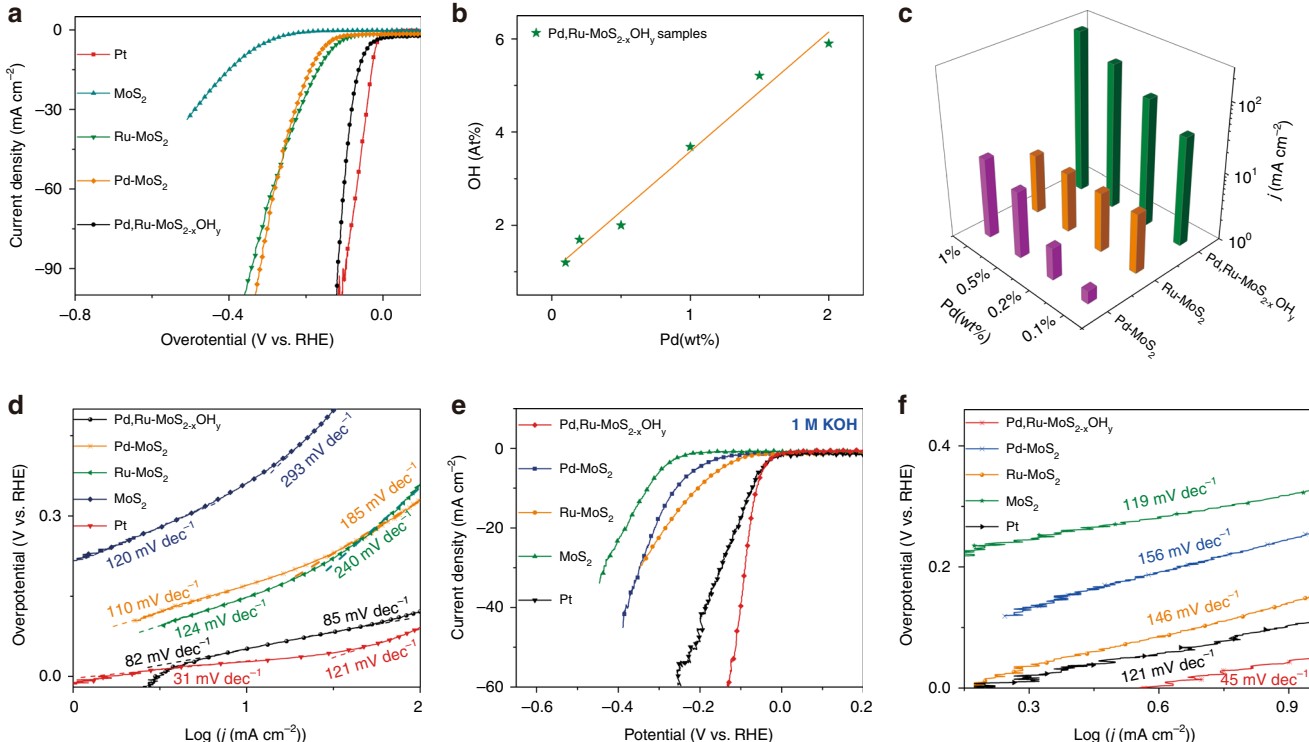

**Fig. 2 Superior activity and stability of Pd, Ru–Mo$_{2-x}$OH$_y$. a** LSV polarization curves of Pt, MoS$_2$, Ru–MoS$_2$, Pd–MoS$_2$ and Pd, Ru–MoS$_{2-x}$OH$_y$ in 0.5 M H$_2$SO$_4$. (with *iR* correction). **b** OH content as a function of Pd content in Pd, Ru–MoS$_{2-x}$OH$_y$ **c** Current density achieved at $\eta = 140$ mV as a function of Pd content in Pd, Ru–MoS$_{2-x}$OH$_y$, Ru–MoS$_2$, and Pd–MoS$_2$. **d** Tafel plots derived from the results given in **a**. **e** LSV polarization curves of Pt, MoS$_2$, Ru–MoS$_2$, Pd–MoS$_2$ and Pd, Ru–MoS$_{2-x}$OH$_y$. (with *iR* correction) in 1 M KOH. **f** Tafel plots derived from the results given in **e**.

(1.55–14.94 mA cm$^{-2}$) and Ru–MoS$_2$ (8.7 mA cm$^{-2}$) by 29.94–223.14 mA cm$^{-2}$, where we attribute the net increase of 193.20 mA cm$^{-2}$ to the increase of the −OH content in the sample. We further support our argument through Tafel slope evaluation (Fig. 2d). In low current density region (within 1–10 mA cm$^{-2}$), the Pd, Ru–MoS$_{2-x}$OH$_y$ (82 mV dec$^{-1}$) exhibits much lower value than Pd–MoS$_2$ (110 mV dec$^{-1}$) and Ru–MoS$_2$ (124 mV dec$^{-1}$), possibly originated from both facilitation of the H* adsorption and double layer structure alternation induced by −OH. In high current density regions, the difference is more evident, where the mass transportation plays a more important role.[34,35] Assuming that the di-anionic surface benefits the hydronium ion transfer, the much smaller Tafel slope (85 mV dec$^{-1}$) of Pd, Ru–MoS$_{2-x}$OH$_y$ than the counterpart Pd–MoS$_2$ (185 mV dec$^{-1}$) and Ru–MoS$_2$ (240 mV dec$^{-1}$) is easily understandable.

It is apparent that HER in alkaline media is much more challenging than that in acidic media, due to the rigorous requirement for additional water dissociation.[19,36,37] The corresponding HER polarization curves of all samples in 1 M KOH are illustrated in Fig. 2e. It is shown that the MoS$_2$ (−332 mV @10 mA cm$^{-2}$), Ru–MoS$_2$ (−210 mV @ 10 mA cm$^{-2}$) and Pd–MoS$_2$ (−260 mV @10 mA cm$^{-2}$) electrodes indeed show limited catalytic performance in alkaline media due to the low water dissociation kinetics. In contrast, the Pd, Ru–MoS$_{2-x}$OH$_y$ electrode ($\eta = 48$ mV@ 10 mA cm$^{-2}$, 131 mV@ 50 mA cm$^{-2}$) shows a breakthrough in the catalytic activity, which far exceeds that of the Pt electrode ($\eta = 75$ mV @ 10 mA cm$^{-2}$, 218 mV @ 50 mA cm$^{-2}$). This corresponds to the highest performance ever reported for MoS$_2$-based catalysts in alkaline media.[10,38,39] The Pd, Ru–MoS$_{2-x}$OH$_y$ catalyst also exhibits more than one magnitude increase in intrinsic activity, evidenced by its superior TOF (Supplementary Fig. 30). To

understand the origin of the differences in the overall catalytic performance between Pd, Ru–MoS$_{2-x}$OH$_y$, and counterpart Ru–MoS$_2$ electrodes, we estimated their relative electrochemically active surface areas using cyclic voltammetry measurements by extracting the double-layer capacitance ($C_{dl}$) (Supplementary Fig. 31). The relative electrochemical active surface areas for the Pd, Ru–MoS$_{2-x}$OH$_y$ is similar to that of Ru–MoS$_2$, indicating that the higher catalytic activity of Pd, Ru–MoS$_{2-x}$OH$_y$ achieved is not due to the increase in surface area. Tafel slope (Fig. 2f) evaluation clearly shows an alternation in reaction mechanism due to the −OH incorporation. Tafel slopes span over 140–160 mV dec$^{-1}$ for the pristine MoS$_2$, Ru–MoS$_2$, and Pd–MoS$_2$, signifying water discharge and formation of H* (Volmer reaction) as the rate-limiting step (RDS). Nevertheless, the Pd, Ru–MoS$_{2-x}$OH$_y$ catalyst represents a Tafel slope at 45 mV dec$^{-1}$, suggesting the overcome of water dissociation barrier and the shift of RDS to the electrochemical desorption of hydrogen (the Heyrovsky step). Thus, we deduce that the exceptional alkaline HER activity of Pd, Ru–MoS$_{2-x}$OH$_y$ is mainly aroused from the OH sites in attracting H$_2$O to the surface and the M–OH sites in facilitating the HO–H bond cleavage, considering Pd–MoS$_2$ and Ru–MoS$_2$ do not exhibit such activity.

We then carried out the long-term operating stability test, where the Pd, Ru–MoS$_{2-x}$OH$_y$ show its very robust nature. Supplementary Fig. 32a, 33 show no observable decay during a 10 h test in both acidic and alkaline medium. The stability of the catalyst in acidic solution is primarily concerned, as the Ru–OH is suspicious to dissolution under the attack of hydronium ions. We therefore carried out a prolonged test for 100 h, where results show an overall decay of only 16 mV (Supplementary Fig. 32b), even comparable to the Pt based catalysts. (Supplementary Fig. 34, see the Supporting Information for details[40]). The OH contents of Pd, Ru–MoS$_{2-x}$OH$_y$ before and after were probed by XPS, and the values are not

obviously changed (Supplementary Fig. 35). Further, no leaching of Ru element in the electrolyte after tests was monitored. We conducted further XPS and (in situ) XANES investigations to examine the valance state of Ru the Pd, Ru−MoS$_{2-x}$OH$_y$ sample after electrolysis. Notably, neither the content nor the valance state of Ru was altered for the post test sample according to the XPS, suggesting that Ru is firmly integrated into the MoS$_2$ backbone and highly stable under electrolytic conditions (Supplementary Fig. 36). Meanwhile, The Ru L$_3$-edge XANES results (Supplementary Fig. 37a) demonstrate no change in the white line resonance strength in comparison to the Pd, Ru−MoS$_{2-x}$OH$_y$ sample before the electrolysis test, thereby suggesting that the average valence of Ru sites is not changed. More importantly, operando X-ray absorption near-edge structure (XANES) provides the most direct evidence to unveil that Ru is not reduced under HER conditions. Supplementary Fig. 38a, b presents the operando XANES spectra at the Ru K-edge of the Pd, Ru-MoS$_{2-x}$OH$_y$ catalyst recorded at different applied potentials. The ex-situ sample, the sample at open-circuit potential, and the ones under cathodic potentials between 0 and −0.05 V all show similar absorption edge, suggesting that Ru maintains its +3 valence state during HER process. Meanwhile, the first derivative of the adsorption edge shows no variation in intensity maximum, thus further suggesting the unaltered valence state of Ru during HER. This result clearly demonstrates that the Ru−OH is stably introduced, where the −OH is chemical stabilized by 1 Ru and 2 Mo atoms and Ru is stabilized by an overall 5.5 covalent S/OH bonds. Further, the post XPS and XANES results (Supplementary Figs. 37b, 39) of the Pd, Ru−MoS$_{2-x}$OH$_y$ sample after electrolysis reveal that Pd is also firmly integrated into the MoS$_2$ backbone and highly stable under electrolytic conditions. Operando XANES spectra (Supplementary Fig. 38c, d) of the Pd K-edge shows no shift of the absorption energy edge of the Pd, Ru−MoS$_{2-x}$OH$_y$ between in situ and ex situ sample, thus implying that no change in the Pd average valence state occurs during the HER process.

### Fundamental understanding of the di-anionic effect

Up to this point, we have demonstrated that di-anionic surface vastly promotes the HER activity in both acidic and alkaline medium. In what follows, we try to obtain fundamental understanding of the di-anionic effect using a combination of theoretical and experimental techniques. We have demonstrated in a previous paper that the Pd bonded S* exhibit $\Delta G_H$ at –0.02 eV[8], which shows optimized chemical adsorption behavior. (see Supplementary Note 2 for details and Supplementary Fig. 36–41). Therefore, we here focus on the effect of –OH part for further performance enhancement. First, we computed interaction between H$^+$(H$_2$O)$_n$ and the surface −OH sites using density functional theoretical calculations. The projected density of states (pDOS) (Supplementary Fig. 42) were performed to study the bonding and electronic structure between –OH and H$_2$O. The delocalized molecular orbitals of O adsorbed on Pd, Ru−MoS$_{2-x}$OH$_y$ interact weakly with the H 1s orbital in the −9.2 to –9.0 eV energy zone, thus confirms the presence of the non-covalent bonding[41,42]. We then carried out electron localization function (ELF) evaluations (Fig. 3a) to measure the excess kinetic energy density due to the Pauli repulsion[41]. The topological image shows that the V(O, H) basin belongs to the OH in Pd, Ru−MoS$_{2-x}$OH$_y$ valence shell sharing a boundary with V(O) basin, typical for hydrogen bonding[43–45]. Further energy evaluation (Fig. 3b) shows a stabilization energy of 58 kJ mol$^{-1}$ (0.58 eV), indicates a hydrogen bonding that is even stronger than the inter-molecular HB energy in water (42 kJ mol$^{-1}$, 0.42 eV)[11]. Thus, the hydronium ions and water molecules (Supplementary Fig. 43 and Supplementary discussion 3) are more easily dragged to the catalyst surface due

to the energy preference. Second, we used Bader charge analysis (Fig. 3c) to understand the reason for the formation of stronger HB. We found −OH sites display strong acceptor-type behavior, characterized by –1.12 e negative charge, almost three times higher than the charge on S atoms (−0.37 ~ −0.58 e)[13]. Moreover, the presence of OH sites brings about local charge redistribution, where the electrons in the MoS$_2$ backbone are drawn to −OH and its adjacent site, thus enormously facilitates the formation of stronger hydrogen bonding due to this electron transfer process. This is more clearly reflected in charge density difference evaluation (Supplementary Fig. 44), where the localized negative charge is obviously focused on OH group and it's nearby S atoms[22]. Third, the hydrogen bonding formation is probed experimentally by the contact angel and the potential of zero charge (PZC) evaluations. Macroscopically, the Pd-MoS$_2$ and Ru-MoS$_2$ samples show contact angels at 125° and 130°, clearly demonstrate their water repelling feature. The Pd, Ru−MoS$_{2-x}$(OH)$_y$ di-anionic surface, on the contrary, shows a water withdrawing feature with a contact angel of 75° (Supplementary Fig. 45). This is further reflected in $E_{PZC}$ (Fig. 3d), where the more polarized Pd, Ru−MoS$_{2-x}$(OH)$_y$ surface represents a much more positive value at 0.550 V vs. RHE in comparison to its counterpart MoS$_2$ (0.095 V), Ru−MoS$_2$ (0.102 V), and Pd−MoS$_2$ (0.318 V)[46]. Thus, the electrode has a strong tendency to drag hydronium ions to its surface and transfer electrons to the reactant, with a lower work function (3.22 eV in comparison to 3.82 eV) confirmed by the ultraviolet photoelectron spectroscopy (UPS)[26] (Supplementary Fig. 46).

The aid of −OH functionality in promoting HER on the di-anionic MoS$_{2-x}$(OH)$_y$ electrode is then proposed in Fig. 4. Without the −OH functionalities, water or hydronium ions are located at the outer Helmholtz plane (OHP) in its clustered structure (Fig. 4a). Thus, extra polarization energy is necessitated to counter balance the HB and brings the reactant from OHP to IHP. In terms of di-anionic MoS$_{2-x}$(OH)$_y$ electrode, however, with much stronger HB formed, water/hydronium ions are preferentially attracted to IHP and leads to further reaction (Fig. 4b, step I$_1$ and step I$_2$). In acidic solution, the protons migrate to the energetically preferred Pd–S* sites (step II$_1$) (the asterisk denotes the adsorption site) and are reduced by one electron to produce Pd–S*–H (step III$_1$) (see Supplementary Note 2 for details)[8]. Finally, another proton from an adjacent H$^+$(H$_2$O)$_n$ reacts with the first $H_{ad}$ to generate H$_2$ (step IV$_1$–V$_1$), in a Heyrovsky mechanism. In alkaline medium, we seek the mechanistic origin for the overcome of water dissociation kinetic barriers. At the Ru–OH site, while −OH attracts water to the surface, we propose that the splitting of water occurs at the Ru site, as shown in step II$_2$. This is supported by the appropriate $\Delta E_{OH}$ (−2.8 ~ −2.54 eV) to a second −OH site at the Ru adjacent S vacancy (Supplementary Fig. 47). The moderate $\Delta E_{OH}$ leads to an easy HO–H bond cleavage with formation of adsorbed H* and Ru-(OH)$_2$, followed by the removal of the second −OH site to its original structure(step III$_3$)[47,48]. Therefore, it is believed that OH works in conjunction with adjacent Ru (Ru–OH) to overcome the water dissociation kinetic barrier[49]. We then carried out further experiments to verify the synergistic role of the M−OH (M = Ni, Co) in activating water in the basal plane of MoS$_2$ (see Supplementary Note 3 for details and Supplementary Figs. 48–51). The transition of RDS away from Volmer to Heyrovsky step is noticed in all other catalysts, further verifying that metal sites and −OH groups work coordinately to surmount the water dissociation barrier.

In summary, we confirm the essential role of reactant friendly interface for kinetic advancement of HER, through constructing a di-anionic surface (S and −OH) on MoS$_2$. The controlled molecular

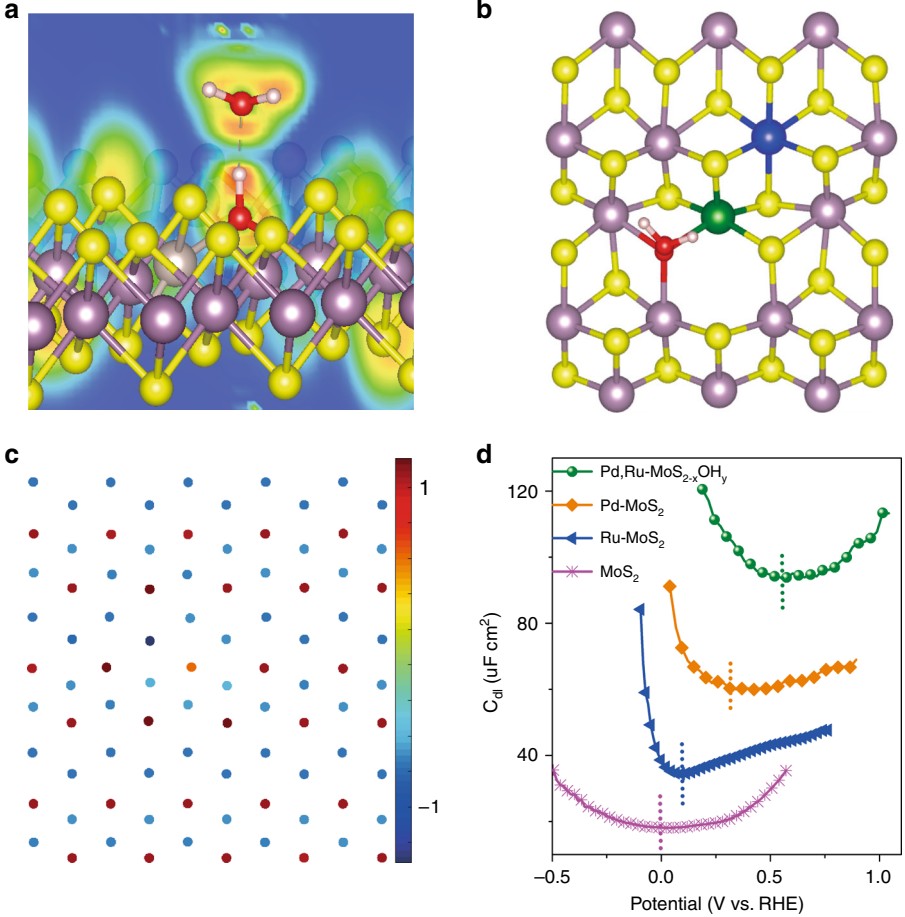

**Fig. 3 Using theoretical and experimental techniques to understanding the di-anionic effect. a** Electron localization function (ELF) evaluations. **b** Interaction energy between $H_2O$ and the surface –OH sites. **c** Bader charge analysis. **d** $E_{PZC}$ measured for Pd, Ru–$MoS_{2-x}OH_y$, Ru–$MoS_2$, Pd–$MoS_2$, and $MoS_2$.

substitution of S by −OH sites was achieved by a sequential element substitution strategy and confirmed by a combination of various techniques. The −OH groups were revealed to form strong non-covalent hydrogen bonding (56 kJ mol⁻¹) to the reactants, thereby attracting the reactants (hydronium ions and water molecules) closer to the inner Helmholtz plane (IHP). Besides, the di-anionic surface is endowed with a water dissociation feature to surmount the Volmer reaction barriers (in alkaline media). Excitingly, the final di-anionic $MoS_{2-x}(OH)_y$ exhibits highest kinetic performance among the existed $MoS_2$ based materials, which is even comparable to (in acid) or better than (in alkaline) the state-of-the-art platinum catalysts, accompanied with outstanding long-term operation stability beyond 100 h. Our work provides a direction for manipulating catalytic properties beyond $MoS_2$ and for heterogeneous catalysis beyond HER via designing catalytic interfaces that are reactants benign.

## Methods

**Materials synthesis**. The Ru–$MoS_2$ material was synthesized through a sol-vothermal method. Firstly, 0.538 g of sodium molybdate ($Na_2MoO_4 \cdot 2H_2O$), 0.6 g of thiourea ($CH_4N_2S$) and 1.094 ml $RuCl_3$ (11.082 mg ml⁻¹) were dissolved in 23.906 ml of water in a beaker and then sonicated for 30 min. The resulting homogenous solution was transferred into a 50 ml Teflon-lined stainless-steel autoclave and heated to 200 °C for 24 h. After cooling to room temperature, the precipitate was separated by centrifugation and washed with methanol and finally dried at 50 °C for overnight. The Pd, Ru–$MoS_{2-x}OH_y$ catalyst was synthesized as follows. Sixty milligram of Ru–$MoS_2$ powder was mixed with 50 ml $H_2O$ in a round-bottom flask, and the mixture was ultrasonicated for 1 h; then the Pd $(OAc)_2$ solution was added and heated to 90 °C for 12 h. The obtained product was obtained by filtration of the suspension, followed by dialysis in deionized water.

**Materials characterization**. Ru and Pd K-edge X-ray absorption spectra were performed at the BL14W1 beamline of the Shanghai Synchrotron Radiation Facility, operating at 3.5 GeV with injection currents of 140–210 mA. Si (111) and Si (311) double-crystal monochromators were used to reduce the harmonic com-ponent of the monochrome beam. The Ru $L_3$-edge XANES spectra were tested at the 4B7A beamline of the Beijing Synchrotron Radiation Facility (BSRF), China, in total electron yield (TEY) mode, where the sample drain current was collected under pressure smaller than $5 \times 10^{-8}$ Pa. while Pd $L_3$-edge spectra were measured by partial fluorescence yield (PFY) mode with a SDD detector vertical to incident monochromatic X-ray. The beam from a bending magnet was monochromatic with a varied line-spacing plane grating and was refocused by a toroidal mirror. The photoluminescence (PL) spectra were performed at room temperature under ambient conditions, using a 532 nm excitation laser. X-ray photoelectron spec-troscopy (XPS) measurements were carried out on Mg Kα radiation source (Kratos XSAM-800 spectrometer).The bulk compositions were evaluated by inductively coupled plasma optical emission spectrometer (X Series 2, Thermo Scientific USA). Nuclear magnetic resonance (NMR) measurements were carried out using a Bruker Avance III 500 MHz spectrometer which was equipped with a double tuned 4 mm MAS probe. Transmission electron microscopy (TEM), high resolution transmis-sion electron microscopy (HRTEM), high-annular dark-field scanning transmis-sion electron microscopy (STEM), and element mapping analysis were conducted on Philips TECNAI G2 electron microscope operating at 200 kV. FTIR spectra were collected using a Nicolet 8700 infrared spectrometer with a resolution of 4 cm⁻¹. All samples were mixed with KBr (Sigma-Aldrich) by grinding before being pressed into pellets.

**Electrochemical measurements**. All electrochemical measurements were carried out in a $N_2$-saturated $H_2SO_4$ solution (0.5 M) standard three-electrode setup using Princeton Applied Research. Inks were prepared by ultrasonically dispersing 5 mg of the samples in a suspension containing 50 µl of a Nafion (5 wt%) solution and 950 µl ethanol. The catalysts loading were calculated as approximately 0.357 mg cm⁻². The HER performances were performed in $N_2$-saturated 0.5 M $H_2SO_4$ using the linear sweep voltammetry at a scan rate of 5 mV s⁻¹. All data presented were *iR* corrected,

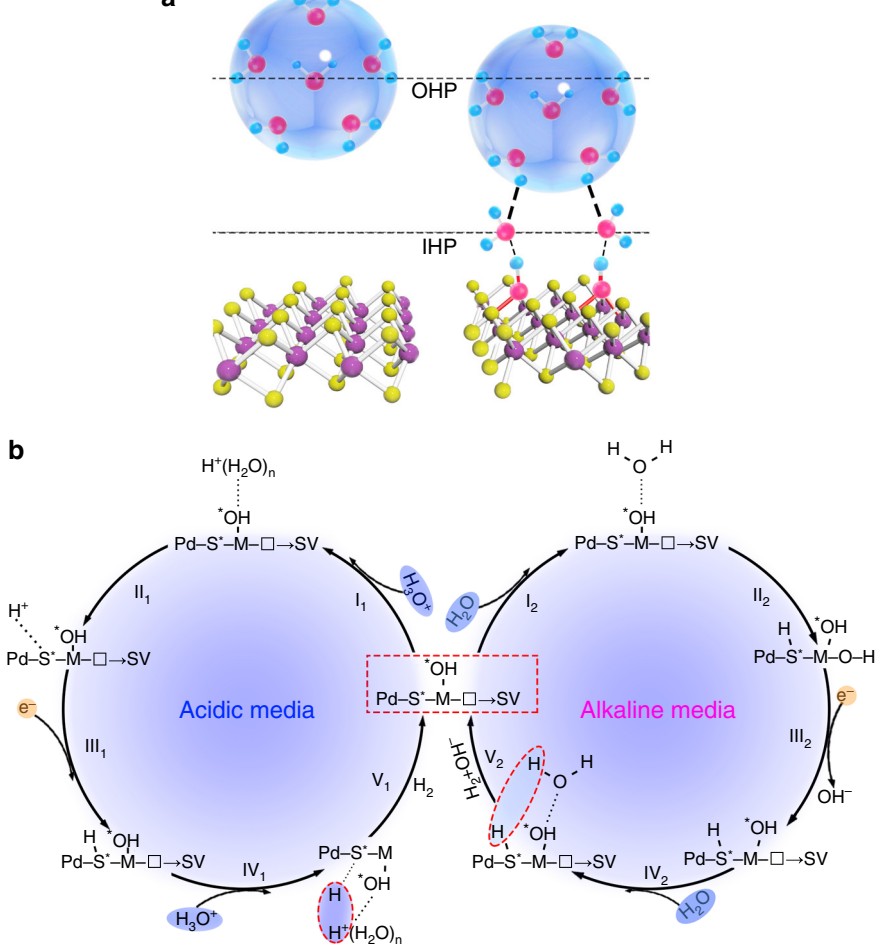

**Fig. 4 Correlation of the HER mechanism with the different reaction condition. a** Schematic illustration of OH functionality in promoting HER on the dianionic $MoS_{2-x}(OH)_y$ electrode through forming strong non-covalent hydrogen bonding to the reactants (hydronium ions or water). **b** Left: HER mechanism of Pd, $Ru–MoS_{2-x}OH_y$ in acidic media; Right: HER mechanism of Pd, $Ru–MoS_{2-x}OH_y$ in alkaline media.

where the solution resistances were determined by EIS experiments. The potential values shown were calibrated vs. the reversible hydrogen electrode (RHE).

**DFT calculations**. Vienna ab initio simulation package of DFT performed accurate to describe the calculated models[50]. The interactions between valence electrons and frozen cores were described by the projected augmented wave method[49]. The GGA method as implemented with Perdew, Burke, and Ernzerhof function[51] was used to describe the exchange-correlation functional component of the Hamiltonian. The kinetic energy cutoff was 400 eV for the plane-wave expansion. To sample the Brillouin zone, the calculation used Monkhorst-Pack $3 \times 3 \times 1$ k-point meshes for structure relaxation and $5 \times 5 \times 1$ k-point grid for the exploration of electronic properties[52]. The geometry convergence tolerance for energy change, max force was $1 \times 10^{-4}$ eV and 0.01 eV Å$^{-1}$, respectively. When building models of different configurations, a large vacuum space of 16 Å was employed to avoid the interactions between $MoS_2$ layers.

## Data availability

The data that support the findings of this study are available from the authors on reasonable request; see author contributions for specific data sets.

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

## Acknowledgements

The work is supported by the National Key R&D Program of China (2018YFB1502400), the National Natural Science Foundation of China (21633008), the Strategic Priority Research Program of CAS (XDA 21090400), Jilin Province Science and Technology Development Program (20190201300JC). Prof. Wei Xing thanks Gusu talent program for the financial support.

## Author contributions

J.G. and Z. Jiang co-supervised the whole work. Z.L., Y.L., and Z. Jin contributed to the synthesis of material and the characterization. Z.L., X.W., Y.L., W.X., J.G., and C.L. contributed to analysis of the electrochemical experiments results. Y.Y. and Z. Jiang contributed to the theory calculation. H.Z. and Z. Jiang contributed to the X-ray absorption fine structure spectroscopy and total electron yield spectroscopy. The manuscript was primarily written by Z.L., J.G., and Z. Jiang. All authors contributed to discussions and manuscript review.

## Competing interests

The authors declare no competing interests.
