## [Peer Review File · Nature Communications]

Reviewers' comments:

Reviewer #1 (Remarks to the Author):

The observation of a particularly high catalytic activity in a modified MoS₂ structure is a potentially interesting finding. The authors propose a structural model and a catalytic mechanism to explain the observed high activity supported by a large number of spectroscopic measurements such as EXAFS, XPS, Raman, FTIR, NMR. While credit is due to the authors for the systematic spectroscopic investigations, I don't think that the proposed structure is unambiguously determined by the provided experimental data, as claimed by the authors. There are a series of omissions they need to clarify before one can judge whether the proposed structure is most likely to be responsible for the observed activity.

1. It would be desirable to provide structural data (SEM, TEM) also on the final material, exhibiting the highest activity, not only the intermediate Ru-MoS₂. For instance, to exclude that the final material is more disordered than the precursor materials, as the increased activity could come alone from the increased disorder (e.g. edge to surface ratio).

2. The authors ignore in their discussions some highly relevant literature that claims the oxygen substitution of S, instead of OH (Nat. Chem. 10, 1246 (2018), Nat. Comm. 10, 3382, (2019)). While I personally think both O and OH substitutions are plausible, the simplistic energetic argument of the authors that OH has a higher binding energy is not sufficient to explain their claim because S vacancies are highly reactive, and they would bind most available species including O, O₂ and OH. What synthesis conditions make OH groups more abundant than other O containing groups for saturating the S vacancies in this case?

3. The fact that OH groups can be identified by FTIR, does not directly imply that they are the substitutional groups. OH groups can be present in various configurations on the sample (e.g. on the edges). The authors only show the presence of OH groups by FTIR on Pd, Ru-MoS₂-xOH_y samples. What about the FTIR spectra of other samples, no OH signature?

4. A rather technical question: in the XPS spectra of Suppl. Fig.18 for supporting the increasing oxygen incorporation with Pd addition, why do the peaks of adsorbed water and surface oxygen increase with increasing Pd content?

I consider that although the manuscript contains a large amount of potentially useful spectroscopic and catalytic data, the proposed structure and therefore catalytic mechanism is not supported well-enough to be suitable for publication in Nature Communications.

Reviewer #2 (Remarks to the Author):

The present work deals with the adaptation of MoS₂ with OH, Pd and Ru for the hydrogen evolution reaction. The work has very interesting points but also weak points. The concept itself reminds of a similar concept introduced by Markovic (DOI: 10.1038/NMAT3313) which is also cited in the here presented manuscript. Here, reactivity trends for the HER were established, indicating an OHad-M interaction in close proximity to a noble metal that dictates the HER reaction rate. In the here described manuscript, Pd and Ru as noble metals are in close proximity to Mo-OH/S. Recently, also the surface oxophilicity was used as parameter for the HER activity (<https://www.sciencedirect.com/science/article/pii/S1388248119300062>) An Cu(I)-OHad---OH₂ activated complex was found to be essential for lowering the energy barrier for water dissociation. A

similar point is shown here. The novelty of the here presented work is not completely clear, the authors also reported a similar paper previously (<https://www.nature.com/articles/s41467-018-04501-4>).

Major points:

It is still not clear what the role of MoS₂ is. The title and abstract suggests that MoS₂ is the catalyst. It is claimed that the performance is the highest ever reported for MoS₂-based catalysts in acidic media. However, MoS₂ is not MoS₂ anymore (“-OH anions molecularly replace S sites at the interface in a controllable manner can create a reactant benign interface.”). It is also questionable if it is really the catalysts. Especially Pd is extremely active towards the HER/HOR; Pd can be even found in some cases on the top of the volcano. Intrinsically very active HER catalysts are used to replace S by OH making it look suspicious that MoS₂/OH should be the real catalyst. The paper suggests that MoS₂ is active (e.g. “We show that the HER activity of the final catalyst exhibits highest kinetic performance exceeding the existed MoS₂ based material in both acidic and alkaline environments.”). Later in the manuscript, it is proposed that Pd-S* is the active specie (“In acidic solution, the protons migrate to the energetically preferred Pd-S* sites (step II1) (the asterisk denotes the adsorption site) and are reduced by one electron to produce Pd-S*-H (step III1). Finally, another proton from an adjacent H+(H₂O)_n reacts with the first Had to generate H₂ (step IV1-V1), in a Heyrovsky mechanism.”). Pd itself is already on top of the volcano with higher current exchange densities than the values reported here, why would the H-adsorption take place on the S* sites? In the previously mentioned paper from Markovic, H* is still adsorbed on the noble metal, not on MO(O)H (DOI: 10.1038/NMAT3313). A control experiment could be to poison the Pd (and Ru) surface and check the activity once more to systematically exclude the possibilities.

A similar point concerns the catalytically active surface area. Different materials are compared with different amounts of noble metals present. From Figure S14a, it is not surprising that the HER activity increases with higher Pd amount. We all know how hard the surface area determination can be. However, please use electrochemical tests to provide estimates of the real surface area (e.g. CO stripping, HUPD, capacitance measurements, redox transitions, etc.), especially of Pd.

The exchange current density lies with 2.884 mA cm⁻² much lower compared to state of the art materials such as Pt (216 mA/cm²Pt;). It is surprising, that the authors claim a comparable activity to Pt despite the large (order of magnitudes) differences.

Additionally, it is advised to measure the exchange current density another time as e.g. proposed by Zheng and coworkers to separate kinetic from mass transport influences (http://jes.ecsdl.org/content/162/14/F1470.abstract?ijkey=03f257db84f70cf8dc5f6285868bb4969884a928&keytype2=tf_ipsecsha)

Please specify the electrochemical reaction conditions: specifically, counter electrode, reference electrode and the material of the electrochemical cell (e.g. glass, Teflon, etc.)

“We therefore carried out a prolonged test for 100 h, where results show an overall decay of only 16 mV (Supplementary Fig. 21b), even better than the Pt based catalysts”

For the OER it was recently shown that bubble blockage of the active surface corresponds mainly to the loss of activity (<http://jes.ecsdl.org/content/166/8/F458.abstract>). The same could be true here as the HER is, similar to the OER, not mass transport limited suggesting that the comparison with bare Pt might not be valid as the surface oxophilicity effects the bubble nucleation and detachment.

“The stability of the catalyst in acidic solution is primarily concerned, as the Ru-OH is suspicious to dissolution under the attack of hydronium ions.”

It is not clear why Ru-OH should be prone to dissolution. Should Ru not be reduced and be thermodynamic stable at these potentials and pH values?

Figure 2: LSVs in KOH: it seems that reductive processes before HER are still taking place, especially in KOH. For Pd/Ru-MoS-OH it seems that the capacitive current is much higher compared to the other samples suggesting that the observed effect might be a simple surface area effect. Especially since the used scan rate was only 5 mV s^{-1} .

Minor points:

It is surprising that the LSVs to these high current densities do not result in bubble formation and that no forced convection was used to remove gas bubbles. Depending on the hydrophilicity/hydrophobicity of the samples, the effective surface is changed drastically, leading to different geometric-based activities. It is suggested to repeat the measurements under rotation to have a defined diffusion layer. The HER specific activities in H₂SO₄ for noble metals is lower compared to non-adsorbing electrolytes such as HClO₄. Please repeat the LSV/CV experiments in HClO₄ to have a fair comparison with bare noble metals.

"Meanwhile, if Ru bonded defects sites (SVs) are formed, they readily captures the nucleophilic -OH species owing to their thermodynamic favorable formation energy of -4.01 eV compared to other species"

This might be true during the synthesis. Why is Ru-OH not reduced under HER conditions?

Terminology. Please make the abbreviation of SV uniform: "is if Ru bonded defects sites (SVs)"; "S vacancies (SVs)"

Reviewer #3 (Remarks to the Author):

Currently, the correlative investigations of electrocatalysts for water splitting process have become one of the hottest areas in view of rather serious energy and environmental crisis. In this work, the authors have performed the detailed experimental and theoretical investigations on the HER catalytic activity of di-anionic MoS₂ material and the related mechanism behind water splitting. This work is interesting. However, the following questions should be addressed before its publication on the Nature Communication:

(1) In this work, the theoretical model related to Ru-doped MoS₂ is adopted to perform the DFT calculation. However, it is not comprehensive considering only the effect of Ru-doping, in view of the fact that doping Pd also has an important effect on the formation of vacancy and the adsorption of OH. Therefore, it is suggested that the co-doping model by Ru and Pd should be constructed to perform the correlative DFT calculations.

(2) According to the authors' description, we can understand that -OH group occupies the vacancy on the surface of Pd,Ru-MoS₂-xOH_y, and brings a high HER catalytic activity. However, it is puzzling why the vacancy (SV) still exists in HER mechanism diagrams of Figure 4. Here, it is suggested that the authors should provide a reasonable explanation for it, as well as the model in Figure 3b.

(3) In this work, the PDOS of O and H is calculated to confirm the existence of non-covalent bonding between OH and H₂O. Here, it's more reasonable that the authors should re-plot this picture by using the data only related to the H and O involved in hydrogen bonding instead of all the H and O atoms.

(4) For the convenience of readers, the authors should add the corresponding energy values in the Supplementary Figure 25. Additionally, the caption of Figure S25a should be corrected since it does not match the corresponding graph.

(5) Moreover, the computational details about ΔG^* in Supplementary Materials should be deleted since the authors do not perform the related calculations.

Reply to Reviewer 1 and revisions made accordingly:

The observation of a particularly high catalytic activity in a modified MoS₂ structure is a potentially interesting finding. The authors propose a structural model and a catalytic mechanism to explain the observed high activity supported by a large number of spectroscopic measurements such as EXAFS, XPS, Raman, FTIR, NMR. While credit is due to the authors for the systematic spectroscopic investigations, I don't think that the proposed structure is unambiguously determined by the provided experimental data, as claimed by the authors. There are a series of omissions they need to clarify before one can judge whether the proposed structure is most likely to be responsible for the observed activity.

Reply: We thank the reviewer for reviewing our work and for the suggestive comments to further improve our manuscript. We have carefully addressed all the comments remarked and our reply to the comments is shown in detail in below.

1. It would be desirable to provide structural data (SEM, TEM) also on the final material, exhibiting the highest activity, not only the intermediate Ru-MoS₂. For instance, to exclude that the final material is more disordered than the precursor materials, as the increased activity could come alone from the increased disorder (e.g. edge to surface ratio).

Reply: Thanks a lot for your constructive comments. We totally agree with the reviewer that increasing the structure disorder (e.g. edge to surface ratio) could increase HER activity of MoS₂-based materials. Based on the reviewer's comment, we have supplemented the structure information with new results to the revised manuscript. The structural data of the final material (Pd,Ru-MoS_{2-x}OH_y) were evaluated by SEM, TEM, XRD, Raman, sub-angstrom resolution aberration-corrected HAADF-STEM microscopy and the nitrogen adsorption/desorption analysis, with results shown in following images in **Figure R1-5**. The SEM and TEM images (**Figure R1-2**) show that Pd,Ru-MoS_{2-x}OH_y is composed of rose-like two-dimensional nanosheets, and the morphology of Ru-MoS₂ is well retained after Pd doping. Additionally, there are no nanoparticles or large clusters appeared in the TEM image, which rules out the possibility of forming palladium sulfide compounds on the surface, in consistent with the X-ray diffraction (XRD) result (no other crystal phase is shown in **Figure R3a**). The Raman spectra of Pd,Ru-MoS_{2-x}OH_y shows similar characteristics with Ru-MoS₂, except for the additional Mo-O bond noticed, indicating that the MoS₂ layered feature in Pd,Ru-MoS_{2-x}OH_y is not changed (**Figure R3b-c**). Moreover, the sub-angstrom resolution high angle annular dark field-scanning transmission electron microscopy (HAADF-STEM) and the selected area electron diffraction (SAED) images (**Figure. R4**) shows an ordered MoS₂ crystalline structure after Pd fixation, which is evident of the atomic dispersion of Pd. Additionally, from the nitrogen adsorption/desorption isotherm (**Figure R5**), we further obtained their specific surface areas (SSA) of 94.380 m² g⁻¹ and 96.941 m² g⁻¹ for Ru-MoS₂ and Pd,Ru-MoS_{2-x}OH_y, respectively. The SSA of Pd,Ru-MoS_{2-x}OH_y hardly deviate from those of the pristine Ru-MoS₂. These experiment results suggest that the structure of the catalysts is well retained after Pd doping, where no significant structure disorder occurs. These results are now added to the supporting information (**Supporting information, Supplementary Figure 10-13, pages 14-17**), and the relevant discussion is added to the manuscript.

Figure R1. a-b SEM of Ru-MoS₂, c-d SEM of Pd,Ru-MoS_{2-x}OH_y.

Figure R2. a-b TEM of Ru-MoS₂, c-d TEM of Pd,Ru-MoS_{2-x}OH_y.

Figure R3. **a** XRD patterns of Ru-MoS₂ and Pd,Ru-MoS_{2-x}OH_y. **b** Raman spectra of Ru-MoS₂. **c** Raman spectra of Pd,Ru-MoS_{2-x}OH_y.

Figure R4. **a** The sub-angstrom resolution aberration-corrected HAADF-STEM images of Ru-MoS₂. **b** The SAED pattern of Ru-MoS₂. **c** The sub-angstrom resolution aberration-corrected HAADF-STEM images of Pd,Ru-MoS_{2-x}OH_y. **d** The SAED pattern of Pd,Ru-MoS_{2-x}OH_y.

Figure R5. a Desorption isotherms of Ru-MoS₂. **b** Desorption isotherms of Pd,Ru-MoS_{2-x}OH_y.

- The authors ignore in their discussions some highly relevant literature that claims the oxygen substitution of S, instead of OH (*Nat. Chem.* 10, 1246 (2018), *Nat. Comm.* 10, 3382, (2019)). While I personally think both O and OH substitutions are plausible, the simplistic energetic argument of the authors that OH has a higher binding energy is not sufficient to explain their claim because S vacancies are highly reactive, and they would bind most available species including O, O₂ and OH. What synthesis conditions make OH groups more abundant than other O containing groups for saturating the S vacancies in this case?

Reply: Thanks for the reviewer's suggestion.

i) Regarding the discussion of literature (*Nat. Chem.* 10, 1246 (2018), *Nat. Comm.* 10, 3382, (2019)):

Thank the reviewer for the constructive suggestion. Indeed, the papers suggested are of high relevancy to the present work. Specifically, the first paper (*Nat. Chem.* 10, 1246 (2018)) confirms for the first time through scanning tunneling microscopy (STM) study that oxygen atoms can spontaneously incorporate into the basal plane of MoS₂ through substitutional oxidation process. The oxygen substitution is reported of high kinetic barrier (~1 eV) and therefore occurs in a very slow rate over a timescale of months, according to transition state theory and the authors' experimental observations, on a single layer MoS₂. In the second paper (*Nat. Comm.* 10, 3382, (2019)), the authors also used single layer MoSe₂ and WS₂ as the model material for study. They combined theoretical calculations with experimental techniques to determine both the atomic structure and electronic properties of an abundant chalcogen-site point defect. They suggest that the point defects are oxygen substitutional defects, rather than vacancies. Both of these previous works are useful references regarding the oxygen substitution on the sulfur sites. Therefore, as the reviewer suggested, we added detailed discussions in our revised manuscript to acknowledge the contributions of these two previous papers and to provide a clearer knowledge status to the readers. As for the difference between our manuscript and these two previous papers, we would like to emphasize that: i) We show that the S atoms are replaced by OH, rather than O in our manuscript due to the high affinity of Ru to OH. ii) The substitution of S by OH is induced by chemical doping method, where Pd and Ru co-doping leads to the formation of S vacancies (SVs) and the attach of OH to the SVs near Ru.

This is the first report that the substitution of S by OH can be controlled through chemical doping technique. iii) -OH is used to engineer the surface structure of the catalysts and enormously facilitate HER performance through attracting the reactants to the surface of the catalysts by hydrogen bonding. The detailed identification of the -OH doping is explained in the following.

ii) Regarding the confirmation of OH substitution rather than O

We truly agree with the reviewer that both O and OH substitutions are plausible. Therefore, we took special caution in identifying the oxygen species, and adopted a combination of EXAFS, XPS, FTIR, ^1H SS-NMR, and DFT calculations to identify the oxygen-containing species in the catalysts.

First, we adopted XPS and EXAFS to confirm that oxygen species are truly doped into the MoS_2 substrate by introducing Pd and Ru simultaneously. Specifically, for the pristine MoS_2 , the Ru- MoS_2 , and the Pd- MoS_2 show the absence of lattice oxygen from XPS (**Figure R6**) and the lack of M-O bond from the EXAFS evaluation (Figure 1d and Supplementary Figure 15). On the contrary, only when MoS_2 is co-doped with Pd and Ru can we achieve the substitution of sulfur by oxygen, where we observed both lattice oxygen from XPS and Ru-O bond from EXAFS. However, XPS and EXAFS cannot tell the difference between O and OH, which demands further verification using other techniques.

Second, FTIR spectra were used to identify the specific configuration of O in the sample. The samples were pretreated under vacuum condition to eliminate water in the sample. As shown in the spectra for Pd,Ru- $\text{MoS}_{2-x}\text{OH}_y$, strong signals of O-H vibrational bands are observed with a broad peak at $3,000\text{--}3,500\text{ cm}^{-1}$ signifies the O-H bond stretching vibration and a peak at 1630 cm^{-1} corresponds to O-H bending vibration. On the contrary, the MoS_2 , Ru- MoS_2 and the Pd- MoS_2 sample show the absence of OH signature (**Figure R7**). Therefore, the results confirm that the oxygen species in the sample are present as OH rather than oxygen itself.

Third, we adopted ^1H SS-NMR spectrum to further prove the presence in O-H form. The ^1H SS-NMR spectrum of the Pd,Ru- $\text{MoS}_{2-x}\text{OH}_y$ catalyst shows a peak at 2.802 ppm, which could match a hydrogen atom bonded to an oxygen atom. Here, the peak at 5~7 ppm is attributed to the H_2O adsorption and the peak at 0.126~0.452 ppm belong to the standard sample peak. The counterparts Ru- MoS_2 , MoS_2 and Pd- MoS_2 samples show no significant ^1H NMR peak at 2.802 ppm, which again identify that the oxygen-containing species in the Pd,Ru- $\text{MoS}_{2-x}\text{OH}_y$ catalyst is -OH (**Figure R8**).

Finally, we performed DFT calculations of the formation energies of various oxygen-containing species, such as O, O_2 , and OH substituting S vacancies. We found that if Ru bonded defects sites are formed, they readily capture the nucleophilic -OH species owing to their thermodynamic favorable formation energy of $-3.89 \sim -4.01\text{ eV}$ compared to other oxygen-containing species (oxygen atom at $-2.32 \sim -2.50\text{ eV}$ and O_2 at $-0.81 \sim -1.23\text{ eV}$). Thus, the successful introduction of -OH is confirmed (**Figure R9**).

Figure R6. a High-resolution XPS results (O $1s$ region) of the Pd,Ru-MoS_{2-x}OH_y. b High-resolution XPS results (O $1s$ region) of the Ru-MoS₂. c High-resolution XPS results (O $1s$ region) of the Pd-MoS₂.

Figure R7. a FTIR spectra of Pd,Ru-MoS_{2-x}OH_y. b FTIR spectra of Ru-MoS₂. c FTIR spectra of Pd-MoS₂. d FTIR spectra of MoS₂.

Figure R8. Proton solid-state nuclear magnetic resonance (^1H SS-NMR) spectrum. a ^1H SS-NMR spectra of Pd,Ru-MoS_{2-x}OH_y. **b** ^1H SS-NMR spectra of Ru-MoS₂. **c** ^1H SS-NMR spectra of Pd-MoS₂. **d** ^1H SS-NMR spectra of MoS₂.

Figure R9. DFT calculation. a-b The formation energies of O occupying the Ru bonded defects sites in Pd,Ru-MoS₂. **c-b** The formation energies of OH occupying the Ru bonded defects sites Pd,Ru-MoS₂. **e-f** The formation energies of O₂ group occupying the Ru bonded defects sites Pd,Ru-MoS₂.

iii) Regarding the preference of OH substitution

We have to mention here that in our sample, the -OH substitution rather than O can mainly be attributed to the high affinity of Ru to OH. It is known that Ru has a high affinity to -OH both theoretically and experimentally. Theoretically, we carried out DFT calculation to provide explanation. The results show that the Ru bonded defects sites (SVs) are more prone to capture the nucleophilic -OH species owing to their thermodynamic favorable formation energy of -4.01 eV compared to other species (oxygen atom at -2.32 eV and O₂ at -1.02 eV) (Supplementary Fig. 2). Experimentally, the phenomenon that Ru is highly affinitive to OH has long been recognized and utilized in catalysis for a variety of reactions such as methanol oxidation and alkaline hydrogen oxidation (in PtRu), where Ru-OH is used as the co-catalyst for CO removal/water dissociation (*Appl Catal B-environ.* 88, 505-514(2009), *J.Phys. Chem. B*, 105,8097-8101(2001), *Langmuir*, 15, 774-779(1999)). In our catalysts, due to the molecular displacement of Mo by Ru, and the in-situ generation of Ru bonded SVs via Pd introduction, the surface doping of -OH is accomplished, as verified by ¹HSS-NMR and FTIR.

We believe there are two reasons for the differed observations between our work and the one mentioned in literature (*Nat. Chem.* 10, 1246 (2018), *Nat. Comm.* 10, 3382, (2019)): 1) In these two references, single layered MoS₂ were both used and therefore the oxidation occurs more easily, although in a very slow rate over a timescale of months, according to the authors' experimental observations. In our work, the oxidation of MoS₂ is more difficult due to the multilayered structure of MoS₂. Therefore, we observe no signal of oxidation on pristine MoS₂, Ru-MoS₂ and the Pd-MoS₂, according to the XPS, EXAFS, ¹HSS-NMR, and FTIR. 2) The high affinity of Ru to OH, which is explained in detail in the above paragraph.

3. The fact that OH groups can be identified by FTIR, does not directly imply that they are the substitutional groups. OH groups can be present in various configurations on the sample (e.g. on the edges). The authors only show the presence of OH groups by FTIR on Pd, Ru-MoS_{2-x}OH_y samples. What about the FTIR spectra of other samples, no OH signature?

Reply: Thanks for the reviewer's suggestion. It is actually a very good question and we agree that ruling out other possibilities are highly challenging. Therefore, we have done a lot of work in verifying the presence of OH. In the manuscript, we used a number of characterization techniques, including extended X-ray absorption fine structures (EXAFS), XPS, PL, FTIR and the proton solid-state nuclear magnetic resonance (¹H SS-NMR) to confirm the substitution of S sites by -OH. First, the nearest-neighbor FT Ru K-edge peak shows obvious shift towards the lower-R position, indicative the emerging of new bond with shorter scattering path. The best fitting analyses clearly show that the path at 2.07 Å is satisfactorily interpreted as Ru-O(H) (the detailed structure of O containing species is not solved here, thus is denoted as -O(H)) contribution. The least-square EXAFS fitting analysis infer the Ru-S and Ru-O(H) coordination numbers of 4.5 (bond length 2.33 Å) and 1 (bond length 2.07 Å), respectively (Supplementary Fig. 7b and Supplementary Table 1). These results clearly demonstrate the incorporation of -O(H) to the sites adjacent to Ru. If -O(H) sites are to replace the S atoms, additional Mo-O(H) should be noticed since -O(H) is to bound with two other adjacent Mo atoms. This is validated by the Mo K-edge EXAFS (Supplementary Fig. 10 and Table 5), where the coordination numbers of Mo-O(H) is found to be

0.3. Meanwhile, through X-ray absorption near-edge spectra (XANES) investigation, the presence of Mo-O(H) is directly confirmed with a shoulder peak observed at 2527 eV in the Mo L₃-edge (Fig. 1e). It is noted that only when Pd and Ru are co-doped through our technique results in the incorporation of the -O(H) into the MoS₂ substrate. The counterparts MoS₂ substrates, Ru-MoS₂, and Pd-MoS₂ were all examined through XAS, where no symbolic signals for -O(H) introduction were noticed. The -O(H) incorporation is further verified by XPS and photoluminescence (PL) spectra. In Pd, Ru-MoS_{2-x}(OH)_y, O 1s peak corresponding to the binding energy of lattice oxygen is clearly observable at 530.5 eV (Supplementary Fig. 11a) in XPS spectra. On the contrary, the Ru-MoS₂ (Supplementary Fig. 11b) and the Pd-MoS₂ (Supplementary Fig. 11c) sample show the absence of the corresponding peak. In PL spectra (Fig. 1f), an obvious peak emerging at 710 nm (1.75 eV) is associated with the -O(H) filled S vacancies, which again only appears when MoS₂ is double doped by Pd and Ru. To here, it is clear that Ru and Pd are both necessary for the -O(H) group introduction. The fact that OH may present on the edge of the sample cannot be ruled out in our sample. However, it is noted that the sample has a very large lateral size (up to micrometer) and therefore the edge sites only contributes to a rather small fraction of the total sites. Therefore, we did not observe the presence of M-O(H) bonds in pristine MoS₂, Ru-MoS₂, and Pd-MoS₂. This is also true for the following FTIR and NMR results.

We used Fourier Transform infrared spectroscopy (FTIR) spectra and the proton solid-state nuclear magnetic resonance (¹H SS-NMR) for final sites verification and see if the structure is O or OH. FTIR spectrum of the Pd, Ru-MoS_{2-x}OH_y sample shows a strong signal of O-H vibrational bands, with a broad peak at 3,000–3,500 cm⁻¹ signifies the O-H bond stretching vibration and a peak at 1630 cm⁻¹ corresponds to O-H bending vibration. On the contrary, the MoS₂, Ru-MoS₂ and the Pd-MoS₂ sample show the absence of OH signature. The ¹H NMR spectrum (**Figure R10a**) of the Pd,Ru-MoS_{2-x}OH_y catalyst shows a peak at 2.802 ppm, which could match a hydrogen atom bonded to an oxygen atom. The counterparts MoS₂ substrates, Ru-MoS₂, and Pd-MoS₂ were all examined through ¹H SS-NMR spectrum, where no symbolic signals for -OH introduction were noticed (**Figure R10c-d**). Thus, these results confirm our claim that the controlled molecular substitution of S by -OH sites was achieved by a sequential element substitution strategy.

We fully agree with the reviewer that the experimental evidence of the FTIR spectra of other counterparts MoS₂, Ru-MoS₂, and Pd-MoS₂ samples is very much useful. According to the reviewer's suggestion, we have added the FTIR spectra of MoS₂, Pd-MoS₂ and Ru-MoS₂ samples in **Figure R11**. FTIR spectrum of the Pd, Ru-MoS_{2-x}OH_y sample shows a strong signal of O-H vibrational bands, with a broad peak at 3,000–3,500 cm⁻¹ signifies the O-H bond stretching vibration and a peak at 1630 cm⁻¹ corresponds to O-H bending vibration. On the contrary, the MoS₂, Ru-MoS₂ and the Pd-MoS₂ sample show the absence of OH signature. These results further support our claim that the controlled molecular substitution of S by -OH sites was achieved by a sequential element substitution strategy. The related contents are now shown in the supplementary Information. (**Supplementary Figure 20-21, Page 24-25**)

Figure R10. Proton solid-state nuclear magnetic resonance (^1H SS-NMR) spectrum. **a** ^1H SS-NMR spectra of $\text{Pd,Ru-MoS}_{2-x}\text{OH}_y$. **b** ^1H SS-NMR spectra of Ru-MoS_2 . **c** ^1H SS-NMR spectra of Pd-MoS_2 . **d** ^1H SS-NMR spectra of MoS_2 .

Figure R11. **a** FTIR spectra of $\text{Pd,Ru-MoS}_{2-x}\text{OH}_y$. **b** FTIR spectra of Ru-MoS_2 . **c** FTIR spectra of Pd-MoS_2 . **d** FTIR spectra of MoS_2 .

4. A rather technical question: in the XPS spectra of Suppl. Fig.18 for supporting the increasing oxygen incorporation with Pd addition, why do the peaks of adsorbed water and surface oxygen increase with increasing Pd content?

Reply: Thanks for the reviewer's suggestive comment. This is indeed a very good question and in the manuscript, we have mentioned that the -OH group endows the interface with reactant dragging functionality through forming strong non-covalent hydrogen bonding to the reactants (hydronium ions or water), using a combination of theoretical and experimental techniques. First, the projected density of states (pDOS) (Supplementary Fig. 24) were performed to study the bonding and electronic structure between -OH and H₂O. The delocalized molecular orbitals of O adsorbed on Pd, Ru-MoS_{2-x}OH_y interact weakly with the H 1s orbital in the -9.2 to -9.0 eV energy zone, thus confirms the presence of the non-covalent bonding. Second, we carried out electron localization function (ELF) evaluations (**Figure R12a**) to measure the excess kinetic energy density due to the Pauli repulsion. The topological image shows that the V(O, H) basin belongs to the OH in Pd,Ru-MoS_{2-x}OH_y valence shell sharing a boundary with V(O) basin, typical for hydrogen bonding. Third, further energy evaluation (**Figure R12b**) shows a stabilization energy of 58 kJ mol⁻¹ (0.58 eV), indicates a hydrogen bonding that is even stronger than the inter-molecular HB energy in water (42 kJ mol⁻¹, 0.42 eV). Forth, the hydrogen bonding formation is probed experimentally by the contact angle and the potential of zero charge (PZC) evaluations. Thus, the -OH group has a strong tendency to attract water molecules to its surface through forming strong hydrogen bonding to water. As monitored by XPS spectra (Fig. 2b, Supplementary Fig. 18 and Table. 9), the -OH content increases with the increase in Pd content, suggesting the essential role of Pd in introducing -OH functional group. This is why the peaks of adsorbed water increase with increasing Pd content.

Additionally, from the Ru EXAFS spectrum, we also found the decrease in overall Ru coordination number from 6.2 to 5.5 after Pd incorporation, corroborating the net creation of Ru adjacent S vacancies in spite of the partial -OH refilling. According to the experiment, we did extra calculation to achieve the Ru-OH formation energy and its influence on the Ru-S bond energy. As shown the **Figure R13**, the formation energy of sulfur vacancy is decreased from 1.79 eV to 1.14 eV after forming the Ru-OH bond. Therefore, this site is more prone to form SV after -OH substitution, which is consistent with our EXAFS data. According to DFT calculations, if Ru bonded defects sites are formed, they readily adsorb additional O containing groups such as O₂, OH, O (**Figure R9** and **Figure R12c**). Therefore, the Ru adjacent S vacancies would bind other available species such as O₂ owing to highly reactive characteristic. From the XPS spectra of Suppl. Fig.18, we can see that the increasing oxygen incorporation with Pd addition. Therefore, the peaks of adsorbed oxygen can also increase with increasing Pd content.

Figure R12. **a** Electron localization function (ELF) evaluations. **b** Interaction energy between H₂O and the surface -OH sites. **c** The formation energies of O₂ group occupying the Ru bonded defects sites Pd,Ru-MoS₂.

Figure R13. Formation energy of S-vacancy.

Reply to Reviewer 2 and revisions made accordingly:

The present work deals with the adaptation of MoS₂ with OH, Pd and Ru for the hydrogen evolution reaction. The work has very interesting points but also weak points. The concept itself reminds of a similar concept introduced by Markovic (DOI: 10.1038/NMAT3313) which is also cited in the here presented manuscript. Here, reactivity trends for the HER were established, indicating an OH_{ad}-M interaction in close proximity to a noble metal that dictates the HER reaction rate. In the here described manuscript, Pd and Ru as noble metals are in close proximity to Mo-OH/S. Recently, also the surface oxophilicity was used as parameter for the HER activity (<https://www.sciencedirect.com/science/article/pii/S1388248119300062>) An Cu(I)-OH_{ad}---OH₂ activated complex was found to be essential for lowering the energy barrier for water dissociation. A similar point is shown here. The novelty of the here presented work is not completely clear, the authors also reported a similar paper previously (<https://www.nature.com/articles/s41467-018-04501-4>).

Reply: We thank the reviewer for the positive remarks and for many constructive suggestions to help us further improve our manuscript.

Before answering the specific questions remarked, we would like to firstly address the question regarding the novelty of the manuscript. We notice that the reviewer mainly questions about the difference of our work with the previously reported work from Prof. Nenad M. Markovic (*Nat.Mater.* DOI: 10.1038/NMAT3313; *Electrochemistry Communications*. 100, 30, 2019) and our own previous work (*Nat Commun.* 9, 2120, 2018). In the following, we will answer the question in sequence:

i)The difference between our work and Prof. Nenad M. Markovic's work.

- Both works from Prof. Markovic dealt the -OH facilitation towards HER/OER in alkaline condition, where the presence of M-OH_{ad} lowers the energy barrier required for splitting water molecule in the HER. In our work, however, we found that the introduction of -OH to the MoS₂ surface also boost the HER activity in acidic medium, where the hydrogen bonding plays a significant role in attracting the hydronium ions to the inner Helmholtz plane (IHP) of the reaction interface.
- In Prof. Markovic's work, both works were focused on the mechanistic side, where the major conclusion is that the formation of M-OH_{ad}-OH₂ is a prerequisite for the lowering of energy barrier required for splitting the water molecule in the HER. The papers did not deal with how to control the surface to provide maximized performance, and the oxides were introduced through either metal oxide clusters (*NATURE MATERIALS* DOI: 10.1038/NMAT3313) or uncontrollable manner (*Electrochemistry Communications* 100 (2019) 30–33). In our work, however, we achieved a reactant friendly di-anion interface, with controlled substitution of S sites by -OH at molecular precision. Therefore, the designed di-anionic MoS_{2-x}(OH)_y catalyst manifests optimized HER activity, showing up-to-date the lowest over potential and highest intrinsic activity among all the MoS₂ based catalysts.

ii)The difference with our own previous work.

The overall rates of most heterogeneous catalytic reactions, including hydrogen evolution reaction (HER), is governed by two fundamentally factors: i) orbital overlap and chemical interactions of adsorbates with the surface sites; ii) the accessibility of reaction interface. In our previous work (*Nature Communication*, **9**, 2120, 2018), we mainly focused on the first part, i.e., the orbital overlap and chemical interactions of adsorbates with the surface sites, by chemically doping MoS₂ with Pd atoms, using a spontaneous interfacial MoS₂/Pd(II) redox technique. In this work, on the other hand, we focus on the second part, namely the accessibility of reaction interface, via introducing molecular -OH anion functionalities to the surface. We found that the -OH functions as an interface modulator, which attracts hydronium ions and water molecules to the inner Helmholtz plane (IHP) and facilitates HER kinetics. Specifically, the following mentioned novelties are all new observations, which make the present manuscript distinct from our previous work:

- The first demonstration of di-anionic MoS₂ surface with controlled substitution of S sites by -OH at molecular precision.
- Revealing the importance of building reactant friendly interface via a combination of experiments and theoretical calculations, where -OH groups are confirmed to drag hydronium ions and water molecules to the inner Helmholtz plane (IHP).
- -OH sites work in conjunction with adjacent metal sites (M-OH) to split water in alkaline medium, thus enormously boosts the HER catalytic behavior.
- The final di-anionic MoS_{2-x}(OH)_y exhibits over potentials of 45 mV and 50 mV, respectively, at 10 mA cm⁻² in acidic media and alkaline media, which far exceeds the performance reported in our previous work and approximates that of Pt.

Major points:

It is still not clear what the role of MoS₂ is. The title and abstract suggests that MoS₂ is the catalyst. It is claimed that the performance is the highest ever reported for MoS₂-based catalysts in acidic media. However, MoS₂ is not MoS₂ anymore (“-OH anions molecularly replace S sites at the interface in a controllable manner can create a reactant benign interface.”). It is also questionable if it is really the catalysts. Especially Pd is extremely active towards the HER/HOR; Pd can be even found in some cases on the top of the volcano. Intrinsically very active HER catalysts are used to replace S by OH making it look suspicious that MoS₂/OH should be the real catalyst. The paper suggests that MoS₂ is active (e.g. “We show that the HER activity of the final catalyst exhibits highest kinetic performance exceeding the existed MoS₂ based material in both acidic and alkaline environments.”). Later in the manuscript, it is proposed that Pd-S* is the active specie (“In acidic solution, the protons migrate to the energetically preferred Pd-S* sites (step III) (the asterisk denotes the adsorption site) and are reduced by one electron to produce Pd-S*-H (step III1). Finally, another proton from an adjacent H⁺(H₂O)_n reacts with the first H_{ad} to generate H₂ (step IV1-V1), in a Heyrovsky mechanism.”). Pd itself is already on top of the volcano with higher current exchange densities than the values reported here, why would the H-adsorption take place on the S* sites? In the previously mentioned paper from Markovic, H* is still adsorbed on the noble metal, not on MO(O)H (DOI: 10.1038/NMAT3313). A control experiment could be to poison the Pd (and Ru) surface and check the activity once more to systematically exclude the

possibilities.

Reply: Thanks for the reviewer's suggestive comments. We have adopted a combination of DFT calculation and experimental techniques to confirm the real active sites and validate that MoS₂, instead of Pd or Ru, is the true catalyst. In addition, according to the reviewer's suggestion, we carried out control experiments using CO poisoning technique to exclude the possibility of Pd/Ru being the active sites. We will show the evidences in sequences:

- Firstly, the Pd-S and Ru-S coordination numbers are 4.5 and 5.5 in the final sample, suggesting a similar coordination environment to Pd/Ru-S₂ and RuS₂ samples. We therefore tested the HER performances of these samples, and the results are shown in **Figure R14**. Notably, neither of the samples shows appreciable HER activity, demonstrating that the Pd and Ru atoms, in their oxidation status, are lacking HER catalytic property in contrast to their metallic states.
- Secondly, the CO poisoning experiments were carried out according to the reviewer's suggestion. **Figure R15** shows the CO poisoning experiment results via bubbling CO into the electrolyte during recording LSVs. Specifically, Pd,Ru-MoS_{2-x}OH_y shows no observable performance decay after CO injection. In contrast, the injection of CO caused huge performance decay of the commercial Pd/C and Ru/C. Thus, it is confirmed that the active sites in Pd,Ru-MoS_{2-x}OH_y are distinct from that of metallic Pd and Ru. Combining the fact that Pd/Ru-S₂ and RuS₂ samples are not active towards HER, we thus claim that the Pd and Ru sites are not the active sites in the Pd,Ru-MoS_{2-x}OH_y material.
- Thirdly, if Pd in the Pd,Ru-MoS_{2-x}OH_y material is the active site, the HER catalytic activity of the material will increase with the increase of Pd content in the sample. However, as shown in **Figure R16**, an optimum value of Pd doping is observed in the Pd-MoS₂, where Pd doping content beyond 10% results in performance decay of the catalysts. This phenomenon can only be interpreted if Pd atoms are not the active sites, who instead function by activating the MoS₂ basal plane through the introduction of sulfur vacancies (SVs) and by activate the adjacent S atoms (Pd-S*-Mo). This is explained in the following through DFT calculations.
- Fourthly, we carried out DFT calculations to probe into the adsorption properties of the materials through calculating the free energy for atomic hydrogen adsorption(ΔG_{H^*}) of different sites. The Pd sites themselves were calculated to be inert as H does not form a very stable adsorption structure on Pd atop site (**Figure R17**). However, the ΔG_{H^*} of the S atop site adjacent to Pd (Pd-S*-Mo) in the Pd-MoS₂ exhibits an almost thermoneutral value of -0.02 eV (**Figure R18**), reaching a ΔG_{H^*} comparable to that of platinum. Therefore, Pd atoms are not the active sites, and instead, they function by activating the Pd bonded sulfur atoms to exhibit optimal ΔG_{H^*} atoms (Pd-S*-Mo).
- Finally, we conducted further XPS and (in situ) XANES investigations to exclude the possibility of generating Pd/Ru metallic particles during the operation. **Figure R19** and **Figure R20-21** show the post XPS and XANES test results of the Pd,Ru-MoS_{2-x}OH_y sample after electrolysis. Notably, neither the content nor the valance state of Pd was altered for the post test sample according to the XPS, suggesting that Pd is firmly integrated into the MoS₂ backbone and highly stable under electrolytic conditions (**Figure R19**). Meanwhile, The Pd L₃-edge XANES results (**Figure R20**) demonstrate no change in the white line resonance

strength in comparison to the Pd,Ru-MoS_{2-x}OH_y sample before the electrolysis test, thereby suggesting that the average valence of Pd sites is not changed. Furthermore, in situ XANES spectra were adopted in the revised manuscript to supplement our claim, as is shown in **Figure R21**. Operando XANES spectra of the Pd K-edge shows no shift of the absorption edge of the Pd,Ru-MoS_{2-x}OH_y between in situ and ex situ sample, thus implying that no change in the Pd average valence state occurs during the HER process. Therefore, in situ XAS provides the most direct evidence to support our claim that Pd will not be reduced under HER conditions.

To here, it is clear that real catalyst is MoS₂, not Pd sites.

Figure R14. The HER activity of RuS₂ and Pd/Ru-S₂.

Figure R15. **a** CO tolerance experiment of the commercial Pd black. **b** CO tolerance experiment

of the commercial Ru/C. **c** CO tolerance experiment of the Pd,Ru-MoS_{2-x}OH_y (0.5 M H₂SO₄). **d** CO tolerance experiment of the Pd,Ru-MoS_{2-x}OH_y (1 M KOH).

Figure R16. a Polarization curves of 2%Pd-MoS₂, 5%Pd-MoS₂, 10%Pd-MoS₂ and 15%Pd-MoS₂. **b** Current densities at overpotential of 0.075 V, 0.1 V and 0.125 V for Pd-MoS₂ with different Pd doping contents.

Figure R17. H atom absorbing at Pd sites of 2H-Pd-MoS₂ and 1T-Pd-MoS₂.

Figure R18. H atom absorbing at S atop site of 1T-Pd-MoS₂.

Figure R19. **a** High-resolution XPS results (Pd 3d region) of the Pd,Ru-MoS_{2-x}OH_y sample. **b** High-resolution XPS results (Pd 3d region) of the Pd,Ru-MoS_{2-x}OH_y after electrolysis (1 M KOH). **c** High-resolution XPS results (Pd 3d region) of the Pd,Ru-MoS_{2-x}OH_y after electrolysis (0.5 M H₂SO₄).

Figure R20. Pd L₃-edge XANES spectra of the Pd,Ru-MoS_{2-x}OH_y samples before and after electrolysis, and the XANES data of the reference standards of Pd foil.

Figure R21. **a** Operando XANES spectra recorded at the Pd K-edge of Pd,Ru-MoS_{2-x}OH_y(in 0.5 M H₂SO₄). **b** The first-derivative of XANES spectra in the left figure (a). **c** Operando XANES spectra recorded at the Pd K-edge of Pd,Ru-MoS_{2-x}OH_y (in 1 M KOH). **d** The first-derivative of XANES spectra in the left figure (c).

A similar point concerns the catalytically active surface area. Different materials are compared with different amounts of noble metals present. From Figure S14a, it is not surprising that the HER activity increases with higher Pd amount. We all know how hard the surface area determination can be. However, please use electrochemical tests to provide estimates of the real surface area (e.g. CO stripping, H_{UPD} , capacitance measurements, redox transitions, etc.), especially of Pd.

Reply: Thanks to the reviewer for the constructive comment. We totally agree with the reviewer that the true active area of metallic Pd is associated with the final performance of the catalysts. Nevertheless, as we mentioned in the answer to the previous question, the Pd atoms are in +2 valence state, and do not directly contribute as the active sites. Meanwhile, the Pd (II) ions are not active towards CO and H adsorption; therefore, we anticipated that no positive signals would be generated through CO stripping and H_{UPD} tests. To validate our anticipation, we carried out H_{UPD} test for the Pd-MoS₂ samples doped with varied amount Pd. The representative CV for H_{UPD} (Figure R22) shows no peak in the H_{UPD} region, demonstrating the absence of metallic Pd. These results again demonstrate that Pd cannot be reduced under HER conditions, revealing that Pd atoms are not the active sites, and instead, they function by activating the Pd bonded sulfur atoms to exhibit optimal ΔG_{H^*} atoms (Pd-S*-Mo). As for the CO stripping and redox transitions experiments, it is not suitable for the present material as MoS₂ initiates its own oxidation (0.75 V) to form MoO₃ before the CO stripping potential (~0.8 V).

Figure R22. a Cyclic voltammograms of 0.1%Pd-MoS₂ electrode in 0.5 mol/L H₂SO₄, scan rate: 50 mV/s. **b** Cyclic voltammograms of 0.2%Pd-MoS₂ electrode in 0.5 mol/L H₂SO₄, scan rate: 50 mV/s. **c** Cyclic voltammograms of 0.5%Pd-MoS₂ electrode in 0.5 mol/L H₂SO₄, scan rate: 50 mV/s. **d** Cyclic voltammograms of 1%Pd-MoS₂ electrode in 0.5 mol/L H₂SO₄, scan rate: 50 mV/s.

The exchange current density lies with 2.884 mA cm⁻² much lower compared to state of the art materials such as Pt (216 mA/cm² Pt). It is surprising, that the authors claim a comparable activity to Pt despite the large (order of magnitudes) differences.

Additionally, it is advised to measure the exchange current density another time as e.g. proposed by Zheng and coworkers to separate kinetic from mass transport influences (http://jes.ecsdl.org/content/162/14/F1470.abstract?ijkey=03f257db84f70cf8dc5f6285868bb4969884a928&keytype2=tf_ipsecsha)

Please specify the electrochemical reaction conditions: specifically, counter electrode, reference electrode and the material of the electrochemical cell (e.g. glass, Teflon, etc.)

Reply: Thanks a lot for your valuable suggestion. This is indeed a very useful comment for us to further improving the quality of the work. After receiving the reviewing comments, we have carefully investigated into how to measuring the HER kinetic accurately. For the HER on Pt, it has been suggested by Prof. Yan (*Journal of The Electrochemical Society*, 162 (14) F1470-F1481 (2015)), Prof. Hubert A. Gasteiger (*Journal of The Electrochemical Society*, 157(11) B1529-B1536 (2010)), and the other references that the RDE method is unable to quantify the HOR activity on Pt in acid due to the combination of mass transport limitation of H₂ and fast reaction kinetics. The measured current is dominated by the rate of H₂ diffusion to the electrode surface, as evidenced by the overlapping HOR polarization curve and concentration over potential curve. Therefore, it's quantification is experimentally difficult, and varied measuring conditions lead to a wide range of experimentally reported values in acid electrolytes at room temperature. For example, HOR exchange current densities measured by the rotating disk electrode RDE technique are reported to be on the order of 1 mA cm⁻²_{Pt} using RDE technique. However, much larger values up to hundreds of mA cm⁻²_{Pt} (such as 216 mA/cm² Pt indicated by the reviewer) have been obtained by microelectrode or hydrogen pump studies. Hence, it is inappropriate to compare the catalytic activity of Pd,Ru-MoS_{2-x}OH_y with Pt using the RDE technique and the claim that Pd,Ru-MoS_{2-x}OH_y has a comparable activity to Pt is not correct. Therefore, we have deleted the statement from the revised manuscript. However, we still have to claim that the Pd,Ru-MoS_{2-x}OH_y catalysts is better than most of the best-characterized MoS₂-based materials in terms of the catalytic activity. As for the influence of mass transportation on the measured performance of Pd,Ru-MoS_{2-x}OH_y itself, we carried out further investigations. As is shown in literature (*Journal of The Electrochemical Society*, 162 (14) F1470-F1481 (2015)), if mass transportation limitation plays a significant role in the performance measurement, there will be a clear rotation speed dependent behavior on the catalysts, as shown in following **Figure R23**. We therefore carried out HER measurement on our Pd,Ru-MoS_{2-x}OH_y catalysts in 0.1 M HClO₄ at different rotation speeds (400 rpm, 900 rpm, 1600 rpm and 2500 rpm). The results are presented in **Figure R24**, which shows minimum influence of H₂ mass transportation on the final catalytic behavior. Therefore, the catalytic performance of the catalyst is believed kinetic controlled on our catalysts.

Meanwhile, the electrochemical reaction conditions are added to the supporting information in detail. And for the convenience of the reviewer, we have also put the added information in the following:

The electrochemical measurements were performed in a glass cell for rotating electrodes (PINE Research Instrumentation), with a reversible hydrogen electrode as the reference electrode, a graphite plate served as the counter electrode and a 5 mm diameter glassy carbon as the working electrode. Cyclic voltammograms (CV) of Pd,Ru-MoS_{2-x}OH_y was recorded between 0.03 to -0.15 V vs. RHE in H₂-saturated electrolytes at a scanning rate of 100 mV s⁻¹. Hydrogen evolution reaction (HER) polarization curves were obtained by RDE measurement in 0.1 M HClO₄ with saturated H₂, at a scanning rate of 10 mV/s and rotation speeds ranging from 400 rpm to 2500 rpm at r.t. (30°C).

Figure R23. Picture from Reference (*Journal of The Electrochemical Society*, 162 (14) F1470-F1481 (2015)) for the HER/HOR behavior on Pt. HOR/HER polarization curves at different rotation speeds (100 rpm to 3600 rpm) on a polycrystalline Pt disk in H₂-saturated 0.1 M HClO₄ at a scanning rate of 10 mV s⁻¹. Dashed gray lines are the diffusion overpotential curves at different rotation speeds.

Figure R24. *iR*-corrected HER polarization curves at different rotation speeds (400 rpm to 2500 rpm) on Pd,Ru-Mo_{2-x}OH_y catalyst in H₂-saturated 0.1 M HClO₄ at a scanning rate of 10 mV s⁻¹.

“We therefore carried out a prolonged test for 100 h, where results show an overall decay of only 16 mV (Supplementary Fig. 21b), even better than the Pt based catalysts” For the OER it was recently shown that bubble blockage of the active surface corresponds mainly to the loss of activity (<http://jes.ecsdl.org/content/166/8/F458.abstract>).The same could be true here as the HER is, similar to the OER, not mass transport limited suggesting that the comparison with bare Pt might not be valid as the surface oxophilicity effects the bubble nucleation and detachment.

Reply: Thanks a lot for your kind reminding. According to the reviewer’s suggestion, we have re-examined the HER catalytic stability of Pt catalyst according to the methods described in the literature (*Journal of The Electrochemical Society*, 166 (8) F458-F464 (2019)) strictly. As shown in **Figure R25**, the Pt exhibits an outstanding long-term operational stability. In the revised manuscript, we have removed the inappropriate description. We also added detailed information about the HER stability measurement of Pt in supporting information as shown below:

The detailed information about HER stability measurement of Pt catalyst as shown below:

All electrochemical measurements (cyclic voltammetry, galvanostatic polarization) were conducted in a conventional three-electrode cell under 30°C using the 750E Bipotentiostat (CH Instruments). A reversible hydrogen electrode (RHE) served as the reference electrode, and a graphite plate served as the counter electrode, respectively. The catalyst ink was prepared by ultrasonically dispersing 1 mg Pt catalyst in a suspension containing 1 mL isopropanol, 10 μ L Nafion (5 wt%) solution and 3.99 mL H₂O, then dropped the ink on a glassy carbon rotating ring-disk electrode (RRDE) with a loading of 20 μ g·cm⁻² to form catalyst film coated electrode by drying in air with 500 rpm rotating. The cyclic voltammetry curves were investigated by continuous potential cycling in H₂-saturated 0.5 M H₂SO₄ solution between 0.1 V and -0.15 V with the scan rate at 0.1 V s⁻¹. And after 200 cycles activation, galvanostatic experiments were carried out directly after the linear polarization curves, whereby a constant current density was applied and the resulting potential was recorded over time.

Figure R25. Galvanostatic experiments of Pt at a current density of 10 mA cm⁻².

“The stability of the catalyst in acidic solution is primarily concerned, as the Ru-OH is suspicious to dissolution under the attack of hydronium ions.” It is not clear why Ru-OH should be prone to dissolution. Should Ru not be reduced and be thermodynamic stable at these potentials and pH values?

Reply: Thanks for the reviewer’s comment. We were concerned about the stability of the -OH in acidic solution, and worried about the dissolution of -OH under the attack of hydronium ions. This phenomenon is similar to the instability of hydroxides in acidic media. In the reviewer’s perspective, the reduction of Ru may also occur due to the negative potential and the low pH. In order to answer the questions remarked by the reviewer, we need to verify the status of Ru during the HER conditions.

- Firstly, we examined the Pd, Ru-MoS_{2-x}OH_y catalyst using XPS characterization after the electrolysis test. As is shown in **Figure R26**, neither the content nor the state of Ru was altered after the electrochemical test, thus suggesting that Ru is firmly integrated into the MoS₂ backbone and have stayed in positive valence state under electrolytic conditions.
- Secondly, the X-ray absorption near-edge structure (XANES) test was further carried out, with the Pd,Ru-MoS_{2-x}OH_y sample before the electrolysis test used as a reference sample. The Ru L₃-edge XANES results (**Figure R27a**) demonstrate no change in the white line resonance strength in comparison to the Pd,Ru-MoS_{2-x}OH_y sample before the electrolysis test, thereby suggesting that the average valence of Ru sites is not changed.
- Thirdly, operando X-ray absorption near-edge structure (XANES) provides the most direct evidence to unveil that Ru is not reduced under HER conditions. **Figure R27b** presents the operando XANES spectra at the Ru K-edge of the Pd, Ru-MoS_{2-x}OH_y catalyst recorded at different applied potentials. The ex situ sample, the sample at open-circuit potential, and the ones under cathodic potentials between 0 and -0.05 V all show similar absorption edge, suggesting that Ru maintains oxidation status during HER process. Meanwhile, the first derivative of the adsorption edge shows no variation in intensity maximum, thus further suggesting the unaltered valence state of Ru during operation.

Therefore, as is already claimed in the answer to the previous question, both Pd and Ru stayed in their oxidation status during the HER reaction, which is due to the strong fixation of these atoms in the MoS₂ substrates.

As for the status of -OH during the reaction, we also probed the OH contents of Pd,Ru-MoS_{2-x}OH_y before and after electrochemical reaction by XPS, from which no obvious alternation in the values are noticed (**Figure R28**). Further, no leaching of Ru element in the electrolyte after tests was monitored. This result clearly demonstrate that the Ru-OH is stably introduced, where the -OH is chemical stabilized by 1Ru and 2Mo atoms and Ru is stabilized by an overall 5.5 covalent S/OH bonds. This is consistent with the stability test in a prolonged constant current test (100 h), where results show an overall decay of only 16 mV (**Figure R29**).

Figure R26. **a** High-resolution XPS results (Ru 2p region) of the Pd,Ru-MoS_{2-x}OH_y before electrolysis at 10 mA/cm² for 10 h. **b** High-resolution XPS results (Ru 2p region) of the Pd,Ru-MoS_{2-x}OH_y after electrolysis at 10 mA/cm² for 10 h.

Figure R27. **a** Ru L₃-edge XANES spectra of the Pd,Ru-MoS_{2-x}OH_y samples before and after electrolysis. **b** Operando XANES spectra recorded at the Ru K-edge of Pd,Ru-MoS_{2-x}OH_y, at different applied voltages from the open-circuit condition to -0.05 V during electrocatalytic HER

in 0.5 M H₂SO₄, and the XANES data of the reference standards of Ru foil. **c** The first-derivative of XANES spectra in the left figure (b).

Figure R28. **a** High-resolution XPS results (O *1s* region) of the Pd,Ru-MoS_{2-x}OH_y/CP samples before and after 5000 cycles. **b** High-resolution XPS results (O *1s* region) of the Pd,Ru-MoS_{2-x}OH_y/CP sample before 5000 cycles. **c** High-resolution XPS results (O *1s* region) of the Pd,Ru-MoS_{2-x}OH_y/CP sample after 5000 cycles.

Figure R29. HER stability in acidic media. **a** Chronoamperometry (CP) tests of Pd,Ru-MoS_{2-x}OH_y and Ru-MoS₂ at a current density of 10 mA cm⁻² (22 h). **b** CP tests of Pd,Ru-MoS_{2-x}OH_y at a current density of 10 mA cm⁻² (100 h).

Figure 2: LSVs in KOH: it seems that reductive processes before HER are still taking place, especially in KOH. For Pd/Ru-MoS_{2-x}OH_y it seems that the capacitive current is much higher compared to the other samples suggesting that the observed effect might be a simple surface area effect. Especially since the used scan rate was only 5 mV s⁻¹.

Reply: We apologize for puzzling the reviewer due to our mistake. By checking our experimental data, we found that the LSV polarization curve of Pd,Ru-MoS_{2-x}OH_y presented for results in alkaline was actually collected at the scan rate of 100 mV s⁻¹. We therefore retested the HER measurement on our Pd,Ru-MoS_{2-x}OH_y catalysts in 1 M KOH at different scan rates (5 mV s⁻¹, 10 mV s⁻¹, 20 mV s⁻¹, 50 mV s⁻¹ and 100 mV s⁻¹). The results are presented in **Figure R30**, which shows the influence of scan rate on the capacitive current. To further verify if the sample is reduced and if the increase in activity is due to surface area effect, we conducted further experiments and the evidences are shown in sequences:

i) Regarding whether the Pd, Ru-MoS_{2-x}OH_y catalyst will be reduced before HER.

Reply: Thanks for the reviewer's comment. In the reviewer's perspective, the reduction of Ru and Pd may occur before the HER conditions in alkaline media. In order to answer the questions remarked by the reviewer, we need to verify the status of Ru/Pd before the HER conditions.

First, we used a home-built cell to perform operando XAFS measurements while following the reaction with a standard electrochemical work station. We used porous carbon cloths as the working electrode for loading catalysts, so that the catalyst could distribute throughout the electrode, substantially in contact with the electrolyte. As a result, it could be ensured that all the catalytic sites probed by X-ray participated in the electrocatalytic reaction. During the measurements, the working electrode potential was decreased in steps from 0 V to -0.1 V versus RHE. The operando XAFS data were collected under the open-circuit condition (immersed in KOH electrolyte), at initial potential (+30 mV) and two representative potentials (-10 and -50 mV). **Figure R31** shows the operando X-ray absorption near-edge structure (XANES) spectra at the Ru and Pd K-edge of the Pd/Ru-MoS_{2-x}OH_y catalyst recorded at different applied potentials. Clearly, the Pd/Ru-MoS_{2-x}OH_y (**Figure R31a**, Pd K-edges) shows no shift in absorption edge between the ex-situ/dry state sample, the sample at open-circuit condition, and the ones under operation potentials(0 mV and -50 mV), thus suggesting that Pd maintains its +2 valence state during HER. A similar phenomenon was observed in the Ru K-edge XANES spectra (**Figure R31b**), thereby suggesting no change of the Ru oxidation state.

Second, we also carried out the ex situ X-ray absorption near-edge structure spectra (XANES) and XPS spectra to examine the Pd, Ru-MoS_{2-x}OH_y catalyst after the electrolysis test in alkaline, with results shown in following images in **Figure R32-33**. The Pd and Ru L₃-edge XANES results (**Figure R32**) demonstrate no change in the white line resonance strength in comparison to the Pd,Ru-MoS_{2-x}OH_y sample before the electrolysis test, thereby suggesting that the average valence of Pd/Ru sites is not changed. Moreover, the XPS spectra (**Figure R33**) show that neither the content nor the state of Pd/Ru were altered after the electrochemical test, thus suggesting that Pd/Ru are firmly integrated into the MoS₂ backbone and have stayed in positive valence state under electrolytic conditions.

Therefore, both Pd and Ru stayed in their oxidation status before/under the HER reaction.

ii) Regarding the surface area effect.

Reply: Thanks a lot for your constructive comments. We agree with the reviewer that increasing the surface area could increase HER activity of MoS₂-based materials. In order to investigate whether or not the increase in HER performance can be attributed to the surface area effect, we first estimated the relative electrochemically active surface area of the samples using cyclic voltammetry measurements through extracting the double-layer capacitance (C_{dl}). We also measured the relative electrochemically active surface area of Ru-MoS₂ catalyst for comparison. As displayed in **Figure R34**, the relative electrochemically active surface areas for the Pd,Ru-MoS_{2-x}OH_y is similar to that of Ru-MoS₂, indicating that the higher catalytic activity of Pd,Ru-MoS_{2-x}OH_y achieved is not due to the increase in surface area. Additionally, from the nitrogen adsorption/desorption isotherm (**Figure R35**), we further obtained their specific surface areas (SSA) of 94.380 m² g⁻¹ and 96.941 m² g⁻¹ for Ru-MoS₂ and Pd,Ru-MoS_{2-x}OH_y, respectively. The SSA of Pd,Ru-MoS_{2-x}OH_y hardly deviate from those of the pristine Ru-MoS₂, which also rules out the possibility of surface area effect.

Therefore, the capacitive current noticed is due to our mistake in using the wrong data at 100 mV s⁻¹. We apologize for our mistake and the data is now corrected in the manuscript. We also rechecked all the electrochemical data to eliminate any similar mistake. We really appreciate the reviewer for the careful inspection and pointing this out.

Figure R30. a LSV polarization curves of Pd, Ru-MoS_{2-x}OH_y in 1 M KOH at scan rate of 5 mV s⁻¹. **b** LSV polarization curves of Pd, Ru-MoS_{2-x}OH_y in 1 M KOH at different scan rates (5 mV s⁻¹, 10 mV s⁻¹, 20 mV s⁻¹, 50 mV s⁻¹ and 100 mV s⁻¹).

Figure R31. **a** Operando XANES spectra recorded at the Pd K-edge of Pd,Ru-MoS_{2-x}OH_y (in 1 M KOH). **b** The first-derivative of XANES spectra in the left figure (a). **c** Operando XANES spectra recorded at the Ru K-edge of Pd,Ru-MoS_{2-x}OH_y (in 1 M KOH). **d** The first-derivative of XANES spectra in the left figure (c).

Figure R32. **a** Ru L₃-edge XANES spectra of the Pd,Ru-MoS_{2-x}OH_y. **b** Pd L₃-edge XANES spectra of the Pd,Ru-MoS_{2-x}OH_y and Pd foil.

Figure R33. **a** High-resolution XPS results (Ru 2p region) of the Pd,Ru-MoS_{2-x}OH_y before electrolysis. **b** High-resolution XPS results (Ru 2p region) of the Pd,Ru-MoS_{2-x}OH_y after electrolysis. **c** High-resolution XPS results (Pd 3d region) of the Pd,Ru-MoS_{2-x}OH_y before electrolysis. **d** High-resolution XPS results (Pd 3d region) of the Pd,Ru-MoS_{2-x}OH_y after electrolysis.

Figure R34. a-b Cyclic voltammograms for Ru-MoS₂ and Pd,Ru-MoS_{2-x}OH_y electrodes at different scan rates. c-d Plot showing the extraction of the double-layer capacitance (C_{dl}) for Ru-MoS₂ and Pd,Ru-MoS_{2-x}OH_y electrodes.

Figure R35. a Desorption isotherms of Ru-MoS₂. b Desorption isotherms of Pd,Ru-MoS_{2-x}OH_y.

Minor points:

It is surprising that the LSVs to these high current densities do not result in bubble formation and that no forced convection was used to remove gas bubbles. Depending on the hydrophilicity/hydrophobicity of the samples, the effective surface is changed drastically, leading to different geometric-based activities. It is suggested to repeat the measurements under rotation to

have a defined diffusion layer. The HER specific activities in H₂SO₄ for noble metals is lower compared to non-adsorbing electrolytes such as HClO₄. Please repeat the LSV experiments in HClO₄ to have a fair comparison with bare noble metals.

Reply: Thanks a lot for your suggestion. We actually carried out electrochemical tests with the electrolyte stirred in the electrochemical cell, which was achieved using a magnetic stirrer. However, as the reviewer commented, such magnetic stir is not able to achieve a defined diffusion layer thickness. We thus carried out new tests using the RDE, which are now complimented in supporting information (**Supporting information, Supplementary Figure 22, page 26**). Furthermore, as suggested by the reviewer, the LSV were tested in 0.1 M HClO₄, and are illustrated in **Figure R36**. The test was performed in H₂-saturated 0.1 M HClO₄ electrolyte with sweep rate of 10 mV s⁻¹ and a rotation of 1600 rpm. Indeed, the performance of catalysts in HClO₄ is significantly enhanced for the commercial Pt/C catalyst.

Figure R36. **a** LSV polarization curves of Pt, MoS₂, Ru-MoS₂, Pd-MoS₂ and Pd,Ru-MoS_{2-x}OH_y in 0.5 M H₂SO₄ (with *i*R correction). **b** LSV polarization curves of Pt, MoS₂, Ru-MoS₂, Pd-MoS₂ and Pd,Ru-MoS_{2-x}OH_y in 0.1 M HClO₄ (with *i*R correction).

“Meanwhile, if Ru bonded defects sites (SVs) are formed, they readily captures the nucleophilic -OH species owing to their thermodynamic favorable formation energy of -4.01 eV compared to other species”

This might be true during the synthesis. Why is Ru-OH not reduced under HER conditions.

Reply:

As clearly suggested from **Figure R15-20** (in reply to the previous questions) measured through XPS, operando XAFS and CO tolerance experiment test, the Ru-OH cannot be reduced under HER conditions. This can be ascribed to two reasons:

- First the Ru-OH is stably introduced, where the -OH is chemical stabilized by 1Ru and 2Mo atoms and Ru is stabilized by an overall 5.5 covalent S/OH bonds. This is validated by the Ru K-edge EXAFS (**Figure R37** and **Table R1**).
- Second, the high Ru-S and Ru-OH bond stability can provide overall structure stability. The

ultralow solubility product of Ru-S_x/Ru-OH_y renders (Table R2-3) the Ru is firmly integrated into the MoS₂ backbone and highly stable under electrolytic conditions. Thus, Ru cannot be reduced under HER conditions.

Figure R37. EXAFS spectra. Fourier transforms of k^2 -weighted Ru K-edge EXAFS spectra of Pd,Ru-MoS_{2-x}OH_y. The blue solid line are experimental results, and the red dotted line are best-fit curves for $R = 1.0\text{--}3.0 \text{ \AA}$, using corresponding $k^2 \chi(k)$ functions in $k = 3.0\text{--}11.3.0 \text{ \AA}^{-1}$.

Table R1. Ru K-edge EXAFS curves fitting parameters.

Sample	path	N	$R(\text{\AA})$	$\sigma^2(10^{-3} \text{\AA}^2)$	$\Delta E_0(\text{eV})$	R -factor
Pd,Ru-MoS _{2-x} OH _y	Ru-S	4.5	2.31	6.6	1.5	0.013
	Ru-O	1.0	2.07	3.0	-1.2	
	Ru-Mo	0.4	2.78	3.0	1.5	

N , coordination number; R , distance between absorber and backscatter atoms; σ^2 , Debye-Waller factor; ΔE_0 , inner potential correction accounting for the difference in the inner potential between the sample and the reference compound.

Table R2. The solubility product constants of the different sulfides.

Molecular formula	Pk_{sp} ($-\lg k_{\text{sp}}$)
α -NiS	18.5
β -NiS	24.0
γ -NiS	25.7

PdS	57.69
FeS	17.2
α -CoS	21.3
β -CoS	25.5
RuS	37

Table R3. The solubility product constants of the different hydroxides.

Molecular formula	Pk_{sp} ($-\lg k_{sp}$)
Co(OH) ₂	14.2
Cu(OH) ₂	19.2
Fe(OH) ₂	15.1
Ni(OH) ₂	14.7
Ru(OH)₂	36

Terminology. **Please make the abbreviation of SV uniform:** “is if Ru bonded defects sites (SVs)”; “S vacancies (SVs)”

Response: Thanks a lot for your suggestion and we are really sorry for our carelessness. We have carefully checked the manuscript and corrected abbreviation for error.

Reply to Reviewer 3 and revisions made accordingly:

Currently, the correlative investigations of electrocatalysts for water splitting process have become one of the hottest areas in view of rather serious energy and environmental crisis. In this work, the authors have performed the detailed experimental and theoretical investigations on the HER catalytic activity of di-anionic MoS₂ material and the related mechanism behind water splitting. This work is interesting. However, the following questions should be addressed before its publication on the Nature Communication:

Reply: We thank the reviewer for the positive remarks and for many constructive suggestions to help us further improve our manuscript.

- (1) In this work, the theoretical model related to Ru-doped MoS₂ is adopted to perform the DFT calculation. However, it is not comprehensive considering only the effect of Ru-doping, in view of the fact that doping Pd also has an important effect on the formation of vacancy and the adsorption of OH. Therefore, it is suggested that the co-doping model by Ru and Pd

should be constructed to perform the correlative DFT calculations.

Reply: Thanks for the reviewer's suggestion. According to the reviewer's suggestion, we have re-calculated the formation energy of sulfur vacancy and the adsorption energy of OH using the co-doping theoretical model by Ru and Pd.

- First, we calculated the energy for the formation of SVs in MoS₂ and Pd,Ru–MoS₂ (**Figure R38**), and the energy for SVs formation decreased by ~1–1.5 eV due to the Pd and Ru co-doping. Thus, we can use the Pd and Ru co-doping strategy to create SVs on the MoS₂.
- Second, the energies of the varied oxygen-containing species (oxygen atoms (O), oxygen molecules (O₂), OH group) on Ru bonded defects sites were further calculated, among the species, OH group is energetically the most favorable in terms of formation energy compared to other oxygen-containing species (**Figure R39-40**). Two different configurations were considered for the calculations, i.e., the Pd and Ru atoms are adjacent to each other (**Figure R39**) and Pd and Ru are not adjacent (**Figure R40**). It is noted that in the latter case, where Pd and Ru atoms do not form Pd-S-Ru bonds, the effect Pd on Ru-O (including O, O₂, OH group) formation energy can be neglected. Therefore, we chose Ru doping model instead of Ru, Pd co-doping model for calculation. As shown in **Figure R39-R40**, both models show the OH doping is the energetically the most favorable sites.

These results are now added to supporting information in revised version of the manuscript. The related content are discussed in the manuscript(**Supporting information, Supplementary Figure 1-4, pages 4-8**).

Figure R38. Formation energy of S-vacancy. Formation of single sulfur defects sites (1SVs) and double sulfur defects sites (2SVs) in Pd,Ru–MoS₂ and MoS₂.

Figure R39. DFT calculation. a-b The formation energies of O occupying the Ru bonded defects sites in Pd,Ru-MoS₂. **c-b** The formation energies of OH occupying the Ru bonded defects sites Pd,Ru-MoS₂. **e-f** The formation energies of O₂ group occupying the Ru bonded defects sites Pd,Ru-MoS₂.

Figure R40. DFT calculation. **a-b** The formation energies of OH occupying the Ru bonded defects sites in Ru-MoS₂. **c-b** The formation energies of O occupying the Ru bonded defects sites Ru-MoS₂. **e-f** The formation energies of O₂ group occupying the Ru bonded defects sites Ru-MoS₂.

- (2) According to the authors' description, we can understand that -OH group occupies the vacancy on the surface of Pd,Ru-MoS_{2-x}OH_y, and brings a high HER catalytic activity. However, it is puzzling why the vacancy (SV) still exists in HER mechanism diagrams of Figure 4. Here, it is suggested that the authors should provide a reasonable explanation for it, as well as the model in Figure 3b.

Reply: We apologized for our confusing message conveyed in this manuscript. We chose the model in Figure 3b based on our experimental and theoretical results.

- Firstly, the least-square Ru EXAFS of fitting analysis infer Ru-S and Ru-OH coordination numbers of 4.5 (bond length 2.33 Å) and 1(bond length 2.07 Å), respectively (**Figure R41** and **Table R4**). These results clearly demonstrate the incorporation of -OH to the sites adjacent to Ru. Meanwhile, the net creation of Ru adjacent S vacancies is observed due to the Pd and Ru co-doping, where the overall Ru coordination number decreases from 6.2 to 5.5 after Pd incorporation. Therefore, experimentally, a net sulfur vacancy adjacent to Ru is observed in spite of the partial -OH refilling.
- Second, according to the experiment, we did extra calculation to achieve the Ru-OH formation energy and its influence on the Ru-S bond energy. As shown the **Figure R42**, the

formation energy of sulfur vacancy is decreased from 1.79 eV to 1.14 eV after forming the Ru-OH bond. Therefore, this site is more prone to form SV after -OH substitution, thereby forming the site structure shown in Figure 3b, which is consistent with our EXAFS data. In the meantime, the Ru-OH bond energy on the new site (-4.00~-4.32 eV) is thermodynamically more stable than the on the single vacancy site (-3.89~-4.01 eV) (**Figure R43**), thereby leading to the final structure shown in Figure 3b.

Figure R41. EXAFS spectra. Fourier transforms of k^2 -weighted Ru K-edge EXAFS spectra of Pd,Ru-MoS_{2-x}OH_y. The blue solid line are experimental results, and the red dotted line are best-fit curves for $R = 1.0\text{--}3.0 \text{ \AA}$, using corresponding $k^2 \chi(k)$ functions in $k = 3.0\text{--}11.3.0 \text{ \AA}^{-1}$.

Table R4. Ru K-edge EXAFS curves fitting parameters.

Sample	path	N	$R(\text{\AA})$	$\sigma^2(10^{-3} \text{ \AA}^2)$	$\Delta E_0(\text{eV})$	R -factor
Pd,Ru-MoS _{2-x} OH _y	Ru-S	4.5	2.31	6.6	1.5	0.013
	Ru-O	1.0	2.07	3.0	-1.2	
	Ru-Mo	0.4	2.78	3.0	1.5	

N , coordination number; R , distance between absorber and backscatter atoms; σ^2 , Debye-Waller factor; ΔE_0 , inner potential correction accounting for the difference in the inner potential between the sample and the reference compound.

Figure R42. Formation energy of S-vacancy.

Figure R43. **a** The formation energies of OH occupying the the Ru bonded defects sites in Ru-MoS₂ with 1SVs. **b** The formation energies of OH group occupying the Ru bonded defects sites in Pd,Ru-MoS₂ with 1SVs. **c** The formation energies of OH occupying the Ru bonded defects sites in Ru-MoS₂ with 2SVs. **d** The formation energies of OH occupying the Ru bonded defects sites in Pd,Ru-MoS₂ with 2SVs. (When Pd and Ru atoms are not coordinated with same S atoms, the effect of Ru and Pd co-doping is similar to Ru-doping, here, we constructed the doping model by Ru to perform the correlative DFT calculations)

- (3) In this work, the PDOS of O and H is calculated to confirm the existence of non-covalent bonding between OH and H₂O. Here, it's more reasonable that the authors should re-plot this picture by using the data only related to the H and O involved in hydrogen bonding instead of all the H and O atoms.

Reply: Thanks a lot for your kind reminder and we apologize for not presenting the calculation details in the manuscript. Actually, we calculated the PDOS of O and H using the data related to the H and O involved in hydrogen bonding instead of all the H and O atoms, we have now added the relevant descriptions in the revised supporting information (**Calculation part, page 71**).

- (4) For the convenience of readers, the authors should add the corresponding energy values in the Supplementary Figure 25. Additionally, the caption of Figure S25a should be corrected since it does not match the corresponding graph.

Reply: Many thanks for the reviewer's comments. We have added corresponding energy values in the Supplementary Figure 25 and for convenience, we also copied it to this response as follows (**Figure R44**):

Figure R44. Water adsorption energies on different sites. **a** The water adsorption energies on the OH sites of Pd,Ru-MoS_{2-x}OH_y. **b** The water adsorption energies on the S sites of Pd,Ru-MoS_{2-x}OH_y. **c** The water adsorption energies on the SVs sites of Pd,Ru-MoS_{2-x}OH_y. **d** The water adsorption energies on the OH sites of Ru-MoS_{2-x}OH_y. **e** The water adsorption energies on the S sites of Ru-MoS_{2-x}OH_y. **f** The water adsorption energies on the SVs sites of Ru-MoS_{2-x}OH_y.

(When Pd and Ru atoms are not coordinated with same S atoms, the effect of Ru and Pd co-doping is similar to Ru-doping, here, we constructed the doping model by Ru to perform the correlative DFT calculations)

(5) Moreover, the computational details about ΔG_{H^*} in Supplementary Materials should be deleted since the authors do not perform the related calculations.

Reply: Thanks a lot for your suggestion and we are really sorry for our carelessness. We have carefully checked the SI's computational details and corrected the related errors.

REVIEWERS' COMMENTS:

Reviewer #2 (Remarks to the Author):

The authors made great efforts to prove that the active sites were MoS₂ based. All questions were answered.

**Concerning Reviewer #1's previous comments:

I went over the comments and for me the authors provided sufficiently proof for their claim. I would never have guessed that it is really the Mo-S sites that are the active sites, but the data seems to be convincing.

There are some minor comments that should be addressed:

There are figures that are neither mentioned in the main nor in the supporting information. Please add the missing cross-linking

For Supplementary Figure 13, the scale bar is missing in all Figures. The SAED pattern does not tell much if not specified which species it is.

The same holds true for S-Figure 12: Please assign the peaks indicated. I could neither find an explanation in the text, the supporting information nor in the figure caption. The same holds true for many figures. If peaks are indicated, they should appear in the main text. Please recheck all figures.

Figure R1-5: The sub-angstrom resolution high angle annular dark field-scanning transmission electron microscopy images do not show any single atoms of Pd or Ru. Is it because the contrast between Mo, Pd and Ru is not sufficient to resolve or is there no Pd/Ru in the shown TEM image? **

Reviewer #3 (Remarks to the Author):

All of my comments have been treated. I would like to recommend the acceptance by Nature Communications. It will be of broad interest to the audience of the journal.

Reviewers' Comments:

Reviewer #2:

The authors made great efforts to prove that the active sites were MoS₂ based. All questions were answered. Concerning Reviewer #1's previous comments: I went over the comments and for me the authors provided sufficiently proof for their claim. I would never have guessed that it is really the Mo-S sites that are the active sites, but the data seems to be convincing.

There are some minor comments that should be addressed:

There are figures that are neither mentioned in the main nor in the supporting information. Please add the missing cross-linking

For Supplementary Figure 13, the scale bar is missing in all Figures. The SAED pattern does not tell much if not specified which species it is.

The same holds true for S-Figure 12: Please assign the peaks indicated. I could neither find an explanation in the text, the supporting information nor in the figure caption. The same holds true for many figures. If peaks are indicated, they should appear in the main text. Please recheck all figures.

Figure R1-5: The sub-angstrom resolution high angle annular dark field-scanning transmission electron microscopy images do not show any single atoms of Pd or Ru. Is it because the contrast between Mo, Pd and Ru is not sufficient to resolve or is there no Pd/Ru in the shown TEM image?*

Reviewer #3:

Remarks to the Author:

All of my comments have been treated. I would like to recommend the acceptance by Nature Communications. It will be of broad interest to the audience of the journal.

Response:

We would like to thank reviewer 2 and reviewer 3 for finding the revision satisfactory and suggesting to accept the paper for publication in Nature Communications. We again deeply appreciate all the referees for their constructive comments and suggestions in the first review, which are all very helpful in improving our manuscript. According to the Editor and reviewer 2's comments, we have revised the manuscript carefully. The issues raised by the reviewer 2 were clearly answered and addressed. The corresponding changes and other corrections were highlighted in yellow color in the revised manuscript. The questions put forward by the reviewers were answered in a point by point manner and shown as follows:

Reviewer #2:

Remarks to the Author:

I went over the comments and for me the authors provided sufficiently proof for their claim. I would never have guessed that it is really the Mo-S sites that are the active sites, but the data seems to be convincing.

Reply: We thank the reviewer for the positive remarks and for the constructive suggestions to help us further improve our manuscript.

There are some minor comments that should be addressed:

1. *There are figures that are neither mentioned in the main nor in the supporting information. Please add the missing cross-linking.*

Reply:

Thanks for the careful inspection; all figures are now mentioned in main or in the supporting information in sequence.

2. *For Supplementary Figure 13, the scale bar is missing in all Figures.*

Reply:

We reprocessed the image of Supplementary Figure 13 according to the suggestion (Figure R1a-b below).

3. *The SAED pattern does not tell much if not specified which species it is.*

Reply:

The selected area electron diffraction (SAED) pattern (Figure 2c) demonstrates that the catalyst displays a well-defined spotted pattern corresponding to the diffraction along the (002) and (100) planes of a 2H phase MoS₂ structure. The relative content has also been added in the supplementary Information.

Figure R1. **a** The sub-angstrom resolution aberration-corrected HAADF-STEM images of Ru-MoS₂, the corresponding scale bar is 1 nm. **b** The SAED pattern of Ru-MoS₂, with the rings corresponding to the expected lattice spacings of MoS₂, representing the (002), (100) reflections for MoS₂, respectively. **c** The sub-angstrom resolution aberration-corrected HAADF-STEM images of Pd,Ru-MoS_{2-x}OH_y, the corresponding scale bar is 1 nm. **d** The SAED pattern of Pd,Ru-MoS_{2-x}OH_y. with the rings corresponding to the expected lattice spacings of MoS₂, representing the (002), (100) reflections for MoS₂, respectively.

4. The same holds true for S-Figure 12: Please assign the peaks indicated. I could neither find an explanation in the text, the supporting information nor in the figure caption. The same holds true for many figures. If peaks are indicated, they should appear in the main text. Please recheck all figures.

Response: Thanks a lot for your careful inspection. We have assigned the peaks indicated of S-Figure 12 in the corresponding figure caption according to the suggestion (Figure R2 below). Moreover, we also have rechecked all figures and assigned the peaks indicated in the revised manuscript.

Figure R2. **a** XRD patterns of Ru-MoS₂ and Pd,Ru-MoS_{2-x}OHy. Four main diffraction peaks around 14 °C, 33 °C, 37 °C and 58 °C are noticed, corresponding to (002), (100), (101) and (110) planes of MoS₂ crystal, respectively. **b-c** Raman spectra of Ru-MoS₂ and Pd,Ru-MoS_{2-x}OHy samples. (The two distinct peaks at 378 cm⁻¹ and 404 cm⁻¹, corresponding to the 2H phase vibrational configurations of the in-plane Mo-S phonon mode (E_{2g}) and the out-of-plane Mo-S mode (A_{1g}); the three extra peaks at ~232,

278 and 333 cm^{-1} , attributable to J_2 , E_{1g} , and J_3 peaks of the 1T phase MoS_2 phonon modes, respectively)

5. *Figure R1-5: The sub-angstrom resolution high angle annular dark field-scanning transmission electron microscopy images do not show any single atoms of Pd or Ru. Is it because the contrast between Mo, Pd and Ru is not sufficient to resolve or is there no Pd/Ru in the shown TEM image?*

Response: As the reviewer commented, Pd, Ru and Mo possess similar Z contrast, therefore, it is hard to distinct Pd and Ru from Mo through sub-angstrom resolution high angle annular dark field-scanning transmission electron microscopy. However, we are sure that no phase segregation occurs due to the heteroatom doping, and the single atom doping is further confirmed by XAS and other techniques, as suggested in the manuscript.